# Developmental adaptations of trypanosome motility to the tsetse fly host environments unravel a multifaceted in vivo microswimmer system

**Sarah Schuster[1], Timothy Krüger[1], Ines Subota[1], Sina Thusek[2], Brice Rotureau[3], Andreas Beilhack[2], Markus Engstler[1]\***

[1]Department of Cell and Developmental Biology, Biocentre, University of Würzburg, Würzburg, Germany; [2]Department of Medicine II, University Hospital Würzburg, Würzburg, Germany; [3]Trypanosome Transmission Group, Trypanosome Cell Biology Unit, Department of Parasites and Insect Vectors, Institut Pasteur and INSERM U1201, Paris, France

**Abstract** The highly motile and versatile protozoan pathogen *Trypanosoma brucei* undergoes a complex life cycle in the tsetse fly. Here we introduce the host insect as an expedient model environment for microswimmer research, as it allows examination of microbial motion within a diversified, secluded and yet microscopically tractable space. During their week-long journey through the different microenvironments of the fly´s interior organs, the incessantly swimming trypanosomes cross various barriers and confined surroundings, with concurrently occurring major changes of parasite cell architecture. Multicolour light sheet fluorescence microscopy provided information about tsetse tissue topology with unprecedented resolution and allowed the first 3D analysis of the infection process. High-speed fluorescence microscopy illuminated the versatile behaviour of trypanosome developmental stages, ranging from solitary motion and near-wall swimming to collective motility in synchronised swarms and in confinement. We correlate the microenvironments and trypanosome morphologies to high-speed motility data, which paves the way for cross-disciplinary microswimmer research in a naturally evolved environment.

**\*For correspondence:** markus. engstler@biozentrum.uni- wuerzburg.de

**Competing interests:** The authors declare that no competing interests exist.

## Introduction

Microswimmers have intrigued the scientific mind since the very first observations of bacteria, protists and spermatozoa (*Dobell, 1932*). More than 300 years ago, Leeuwenhoek not only observed free swimming organisms, but also studied ´animalcules´ from animal and human environments, especially the fate of spermatozoa in the female genital tract (*Leeuwenhoek, 1685*). Still, to the present day, model systems for the study of microswimmers in their natural habitats are few and far between. Spermatozoa are *the* model microswimmers, due to their availability as free swimming cells and their importance for sexual reproduction. The significance of the physical properties of the female genital tract for sperm motility and internal fertilisation success has been recognised (*Fauci and Dillon, 2006*; *Kirkman-Brown and Smith, 2011*), but this system naturally remains challenging for in vivo analysis.

Another major microswimmer model is *E. coli*. The rotating prokaryotic flagellar apparatus has been intensively studied, mainly in order to unravel the mechanisms of chemotaxis (*Micali and Endres, 2016*). Recently, there is great interest in the collective behaviour of prokaryotes and the implications of real life surroundings, as cells seldom stay alone or move without encountering

mechanical obstructions for long periods of time (*Chaban et al., 2015*; *Persat et al., 2015*). In fact, bacterial collectives constitute relevant ecosystems themselves, that is as biofilms or intestinal microbiomes (*Ben-Jacob et al., 2016*).

The third major group of natural microswimmers are the eukaryotic ciliates and flagellates. Several free swimming protists, pre-eminently the green alga *Chlamydomonas*, have been used as model organisms during the last century, for the analysis of phototaxis, ultrastructural characterisation of the axonemal system and analysis of intraflagellar transport, to name only a few important topics (*Simon and Plattner, 2014*). Again, the relevance of these organisms in their natural habitats is only recently being fully appreciated, as integrated efforts to analyse the huge global diversity of free swimming protists on the one hand (*de Vargas et al., 2015*) and to elucidate the mechanisms of unicellular parasites in their host system on the other hand, are under way (*Krüger and Engstler, 2015*).

The life cycles of important parasitic protists, including the agents of malaria, Chaga´s disease and African sleeping sickness have been recognised since the late nineteenth century (*Leadbeater and McCready, 2000*). African trypanosomes were amongst the first blood parasites to be observed '..., wriggling about like tiny eels and swimming from corpuscle to corpuscle, which they seem to seize upon and worry.' (*Bruce et al., 1895*). The flagellates were found in blood, tissues and cerebrospinal fluid of various animal species and humans, and their full developmental cycle, involving transmission by the tsetse fly vector, was subsequently described (summarised in [*Steverding, 2008*]).

Since then, trypanosomes have become important model organisms, due to their medical importance and fascinating cell biology, including distinctive genetic features and multifaceted developmental stages (*Hoare, 1972*; *Matthews, 2015*; *Vickerman et al., 1988*). Although trypanosomes have always fascinated as microswimmers, the exact swimming mechanism of *Trypanosoma brucei* has only recently been elucidated (*Heddergott et al., 2012*). The parasite is unusual among the flagellates, as the greater part of the flagellum is attached to the cell body, winding around it in a helical course. The flagellum produces waves from both ends of the elastic cell body, which let the cells tumble and twist, producing the wriggling or corkscrew-like trypanosome movement, typically observed in culture media or blood smears. Importantly, the mechanical parameters of the surroundings, that is fluid viscosity or presence of obstacles, influence the parasite's motile behaviour, affecting the frequency ratio of bidirectional flagellar beating and inducing persistent unidirectional movement (*Heddergott et al., 2012*). Thus, trypanosomes seem to have evolved to be highly versatile swimmers, adapted to react flexibly to different mechanical properties of various microenvironments. This became clear, when the characteristic motility behaviours of different trypanosome species were analysed under changing physical conditions. The parasites exhibited a species-specific dynamic adjustment of motile behaviour to various physical surroundings, which could correlate with their preferred infection niches within their mammalian hosts (*Bargul et al., 2016*). The importance of specific niches during infection has been recognised and is currently being scrutinised (*Caljon et al., 2016*; *Capewell et al., 2016*; *Trindade et al., 2016*).

As the interest of biological and especially physical research is focusing on collective swimming behaviour and the influence of borders and confinement, accessible and controllable in vitro and in vivo systems are in demand (*Elgeti and Gompper, 2013*). The long-term goal is to pave the way for multidisciplinary explanations of dynamic behaviour in complex living systems. To this effect we describe here the first enclosed host-parasite system that is amenable to highly detailed analysis of diverse microswimmers in defined microenvironments.

Trypanosomes are transmitted to and from their mammalian host by insect vectors. *T. brucei* is taken up by the tsetse fly during a blood meal, whereupon the parasites undergo a complex developmental cycle, while traversing various organs of the tsetse´s alimentary tract (*Ooi and Bastin, 2013*; *Rotureau and Van Den Abbeele, 2013*). The development involves several genetically fixed physiological changes, allowing the adaptation to significantly different host compartments and striking morphological changes, which greatly influence motile behaviour. Motility is necessary for successful infection and transmission back to the mammalian host (*Rotureau et al., 2014*) and might be of paramount importance for passing several 'bottlenecks' in trypanosome development (*Dyer et al., 2013*).

We consider the trypanosome-tsetse system as particularly attractive for studying flagellate microswimmers in their natural habitats. The small size of the insect allows measurements of

trypanosome swimming behaviour at very different scales, ranging from the observation of all parasites in whole flies to single cell analyses with high spatiotemporal resolution. As we show in this work, the system´s motile occupants exhibit all kinds of behaviour posing prevailing questions in microswimmer research on the one hand, and having possible implications for the cell and developmental biology of the parasites on the other hand. This also means potential insight into the evolution of host-microbe systems and infection processes, and therefore further creation of bridges between physical and biological research.

## Results

### Multicolour light sheet fluorescence microscopy reveals the complex three-dimensional architecture of the microswimmer habitats in the tsetse vector

In order to make the trypanosome-tsetse system experimentally accessible, we first detailed the in vivo boundary conditions that could influence the motile behaviour of the different developmental stages of *T. brucei* within the tsetse alimentary tract. For this purpose, we adapted light sheet fluorescence microscopy (LSFM) to map the tsetse fly´s internal topology. This technology allowed us to record high resolution optical sections of complete fly body parts and generate three-dimensional reconstructions of intact tissues. Multicolour LSFM simultaneously localised epithelial tissues via autofluorescence, the peritrophic matrix (PM) through rhodamine-labelled wheat germ agglutinin (WGA) and the infecting trypanosomes by GFP-expression in the nucleus. The PM is a non-cellular, glycosaminoglycan, glycoprotein and chitin containing, cylindrical sleeve, that continuously lines the gut epithelium and acts as a physical barrier for pathogens ingested within the blood meal (*Lehane et al., 1996*; *Lehane, 1997*).

To gain an overview of the fly´s intact digestive tract, the head, legs and wings were removed before preparation of the remaining body for light sheet microscopy by an adapted clearing procedure, which renders the insect cuticle and organs translucent. Stacks of fluorescence images were recorded, scanning the complete abdomen and thorax (*Video 1*). The sustainment of the fly body is apparent in a surface rendering model of the autofluorescent cuticle (*Figure 1A*). The extent of the intestinal lumen and other thoracic regions can be identified in the appropriate slices of the stack (*Video 1*). A single image reveals the posterior midgut lumen surrounded by autofluorescent abdominal tissue and a narrow intestinal channel leading to the anterior part of the midgut in the thorax (*Figure 1B*). The thoracic flight musculature is prominent in the autofluorescence channel.

Next, we surgically removed the intact tsetse digestive tract (*Figure 1C,D*) for higher resolution LSFM recordings. We first visualised parts of the posterior and anterior midgut of uninfected flies, after staining the proteoglycan matrix of the PM with red fluorescent WGA (*Figure 2*, *Video 2*). The intact gut tissue (plus associated structures, e.g. fat bodies) was reconstructed (white in *Figure 2*, top panel). The PM was visualised as a compactly folded tubular structure underlying the gut tissue (cyan in *Figure 2*). After the fly had completed digestion, the PM showed an astonishing degree of convolution in both gut regions, the full extent of which can be appreciated in single slices (*Figure 2*, bottom panel, *Video 2*). The PM was folded extensively, producing a multitude of channels, folds and crevices, thereby increasing the absolute surface area. The diameter and thus the surface area of the sleeve significantly increased towards the posterior part of the midgut (*Figure 2B*), allowing the PM to adapt to the extreme swelling of the gut during a blood meal.

Having attained a good impression of the amazingly complex geometry of the ectoperitrophic space, we simultaneously visualised the trypanosomes that populate, navigate and develop

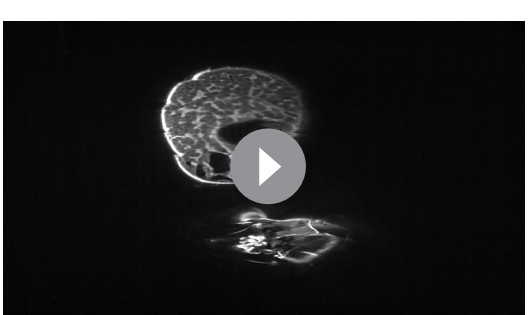

**Video 1.** Original LSFM stack used for visualisation of the tsetse fly in *Figure 1*.

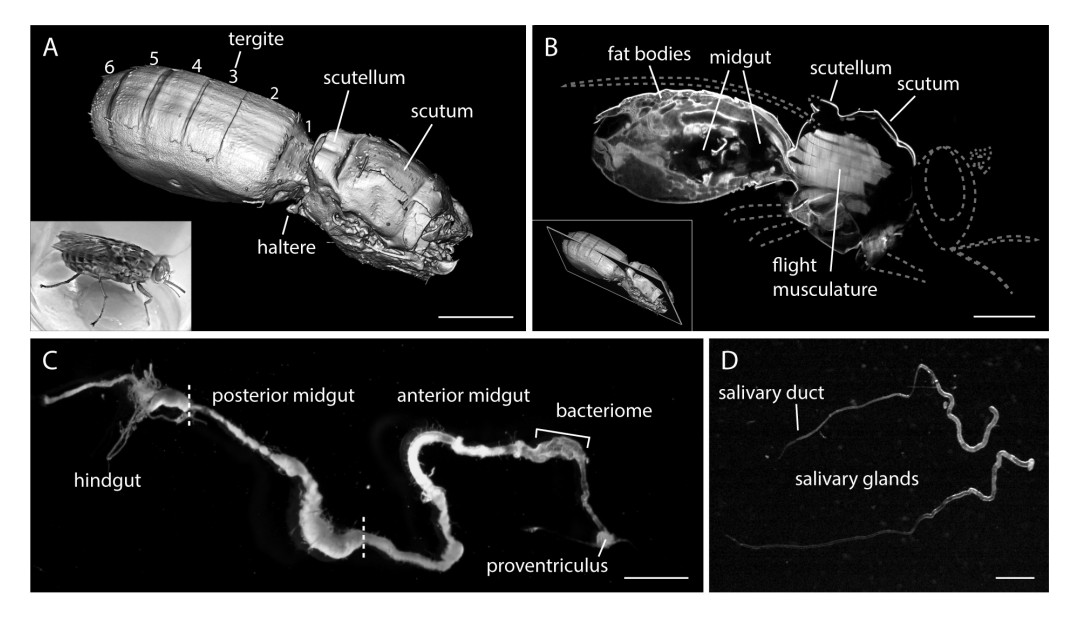

**Figure 1.** The alimentary tract is the trypanosome habitat within the tsetse fly. (**A**) A surface rendering model of an intact female tsetse fly thorax and abdomen, visualised by multicolour LSFM. Head, extremities and a part of the back shield (scutellum) were removed. The fly was fixed, bleached and cleared to enable the autofluorescence recording of the complete abdomen and thorax. The inset shows a living tsetse fly. Scale bar: 1 mm. (**B**) Mid-section of the three-dimensional data set (*Video 1*) showing internal anatomical details. The grey dotted lines indicate removed body parts (not to scale). Inside the thorax the flight musculature is strongly autofluorescent, whereas mainly fat bodies are visible in the abdomen. The abdominal midgut is detected as a void volume of low to negligible autofluorescence. The inset indicates the position in the image stack. Scale bar: 1 mm. (**C**) Surgically removed, intact alimentary tract of a teneral fly. The midgut was sprawled out and freed from remaining tissue. The midgut is divided into a posterior part, where the blood meal is digested, and a thinner anterior part, which includes the bacteriome and ends in the proventriculus. Scale bar: 3 mm. (**D**) Two salivary glands with the thinner salivary ducts. Scale bar: 1 mm.

in this environment. For this purpose, we infected flies with trypanosomes constitutively expressing fluorescent proteins either in the nucleus, the flagellum (EGFP) or the cytoplasm (tdTomato). We confirmed the developmental progress of the fluorescent protein-expressing pleomorphic AnTat1.1 cell lines through all morphological stages, including the final production of mammalian-infective stages (metacyclic form, *Figure 3 - Video 1* and *Figure 3 - Video 2*). In order to generate a map of the characteristic infection process from the posterior midgut to the proventriculus, we analysed these organs, removed from flies at different time points after infection, with trypanosomes express-ing a cytoplasmic fluorescent marker protein (*Figure 3A,B*). The amount and location of trypano-somes was evaluated in defined sub-regions of several digestive tracts, prepared and recorded in an identical manner (*Figure 3B*). A spatiotemporal map of the mean abundance of parasites shows the characteristic establishment of an early posterior midgut infection, the diminishing of populations between days four and six and the invasion of the anterior midgut and the proventriculus from day seven on (*Figure 3A*). After days six to seven, the transmission of the parasites depends on develop-mental stages that cross the PM and colonise the ectoperitrophic space to migrate to the foregut (*Gibson and Bailey, 2003*).

We performed LSFM at early post-infection time-points (1–5 days), when the trypanosomes had adapted to the gut lumen, and late time-points (from 7 days post-infection), after the cells had crossed the PM to infest the ectoperitrophic space (*Figure 3C,D*, *Video 3*, *Figure 3—Video 3*). The LSFM data allowed the clear localisation of single GFP-labelled trypanosome nuclei in the gut lumen of insects early during infection (*Figure 3C*, *Video 3*). Multi-channel volume rendering models showed the trypanosomes evenly distributed, mostly in larger cavities, encased completely by the PM. In contrast, the reconstructed models of later infection stages revealed the nuclei located in the folds of the PM and between the surfaces of PM and epithelium (*Figure 3D*, *Video 3*). The 3D-ren-dering models of the gut tissues (white) and PM (cyan), viewed from the inside of the gut lumen,

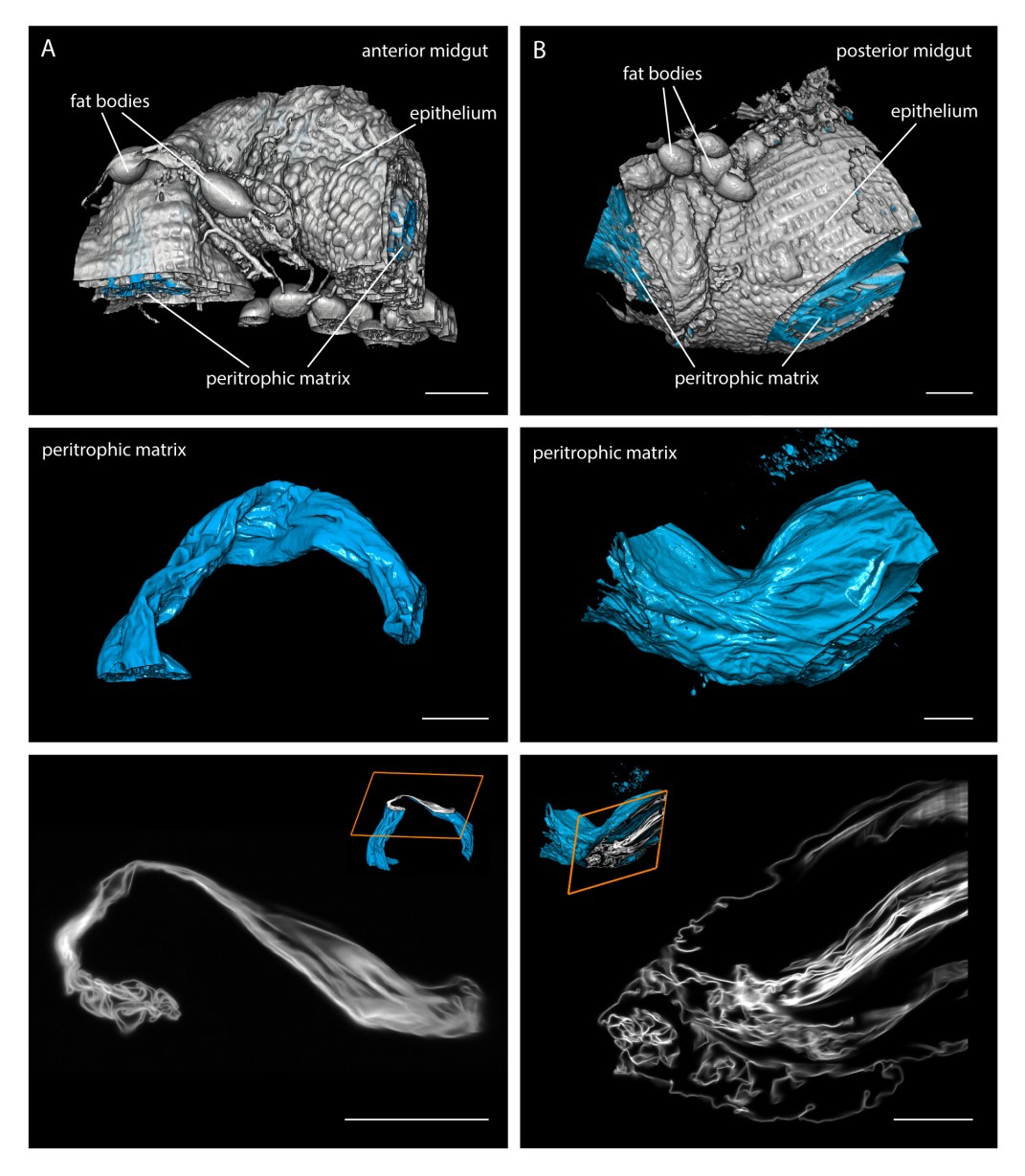

**Figure 2.** Multicolour light sheet fluorescence microscopy details the highly convoluted peritrophic matrix within the tsetse midgut. Fly midguts were surgically removed 1–2 days after the last meal. The intestinal tissue exhibits a strong autofluorescence signal, whereas the PM (cyan) is visualised with rhodamine-labelled WGA. The grey surfaces in the top panel show the epithelial tissue surrounding the PM and attached residual fat bodies. The outer surfaces of the PMs are depicted in the middle panel. The bottom panel shows single image slices of 3D stacks, illustrating the complex membrane folding. The position within the volume stack is shown as orange box in the cutaway model (inset). (**A**) Representative part of the anterior midgut region. (**B**) The posterior midgut region has a larger diameter and contains a more convoluted PM, as well as large void sections of the endoperitrophic gut lumen. Scale bars: 100 μm. *Video 2* contains animated versions of the 3D-PM models and the entire LSFM stacks.

showed trypanosomes (yellow), tightly packed in groups or as single cells, concentrated around the exterior face of the PM in the ectoperitrophic space.

Our results show that LSFM is well suited to determine the in situ parasite population status in body tissues of arbitrarily complex geometries, with single cell resolution (*Figure 3—Video 3*). This becomes even more evident by the analysis of structurally distinct regions of the fly´s alimentary tract, the bacteriome (*Figure 4A*) and the proventriculus (*Figure 4B*, *Video 4*).

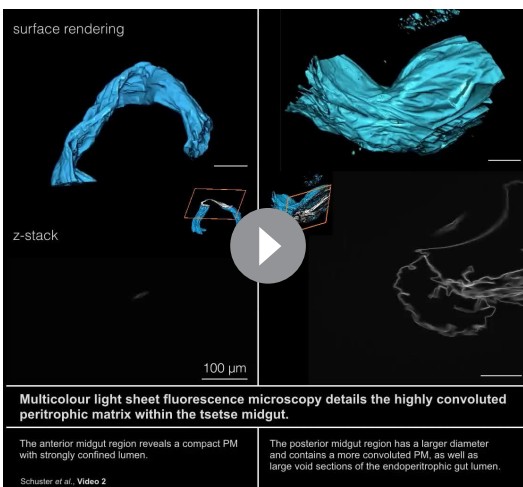

**Video 2.** Multicolour light sheet fluorescence microscopy details the highly convoluted peritrophic matrix within the tsetse midgut.

The bacteriome is a conspicuous region adjacent to the anterior midgut of the fly, containing essential endosymbiotic bacteria, particularly the obligate mutualist *Wigglesworthia*, which resides intracellularly in bacteriocytes (*Aksoy, 1995*; *Wang et al., 2013*). The LSFM analyses showed the distribution of trypanosomes around this area (*Figure 4A*, middle). The images revealed the ectoperitrophic distribution of trypanosomes during late stage infections.

The proventriculus is an intricately shaped organ connecting the anterior midgut, the crop and the salivary duct (*Figure 4B*). The trypanosomes were massively concentrated on the surface of a bent, toroid cellular structure surrounding the entry and exit sites of the proventriculus (*Figure 4B*, lower panels, *Video 4*). The PM is synthesised within these specialised cells and pushed rapidly into the anterior midgut region. As it is unknown if trypanosomes are able to cross the PM here, or how they traverse the proventriculus and continue on their journey to the salivary glands, this region is currently being intensely studied (*Rose et al., 2014*).

The advanced use of LSFM has enabled us to complete the first goal of our study, namely to provide a comprehensive high resolution map of the in vivo topology and boundary conditions of the natural microenvironments that trypanosomes experience on their journey through the tsetse fly. This information is necessary and the resolution is sufficient to interpret any behaviour of swimming trypanosomes acquired by live analyses.

## Three-dimensional morphometry and dynamic cellular waveform analysis of developmental stages characterise specific flagellate microswimmer types

Having detailed the architecture of their tsetse habitat, the next aim was to comparatively analyse the different types of resident trypanosomes, initially in vitro. This should ultimately allow progression to the final step of our endeavour (next section), namely the examination of the trypanosomes' motile behaviour within the fly.

The characterisation of trypanosomes three dimensional cell body shape and the attachment conformation of their force-producing flagella, has been important in elucidating the parasites precise swimming mechanism (*Heddergott et al., 2012*), the characterisation of species-specific differences in motile behaviour (*Bargul et al., 2016*), as well as the numeric simulation of the microswimmers motility in a hydrodynamic surrounding (*Alizadehrad et al., 2015*). The morphometric analysis allows us to assess the chirality and potential hydrodynamic behaviour of each developmental microswimmer type.

As 3D-morphometry with sufficient spatiotemporal resolution is necessarily performed with fixed cells, the data does not allow the measurement of the morphology´s impact on motile behaviour. Therefore, in order to extend the trypanosome morphotype analysis to living cells, we had previously introduced the concept of a dynamic cellular waveform, which describes the propagation of each flagellar wave along the elastic trypanosome cell body and the resulting deformation of the microswimmer along the time axis (*Bargul et al., 2016*). This method yields the information required for understanding the hydrodynamic impact of the flagella-driven, elastic microswimmer body.

We determined the three-dimensional structure of all relevant developmental stages of the pleomorphic AnTat1.1 trypanosomes (*Figure 5*). Trypanosomes were isolated from different organs of flies infected for at least 20 days, their entire cell body and flagellar membrane were fluorescently labelled, their nuclei and kinetoplasts were stained with DAPI and the cells were rapidly fixed. The labelled trypanosomes were microscopically imaged in order to construct 3D-volume models, which

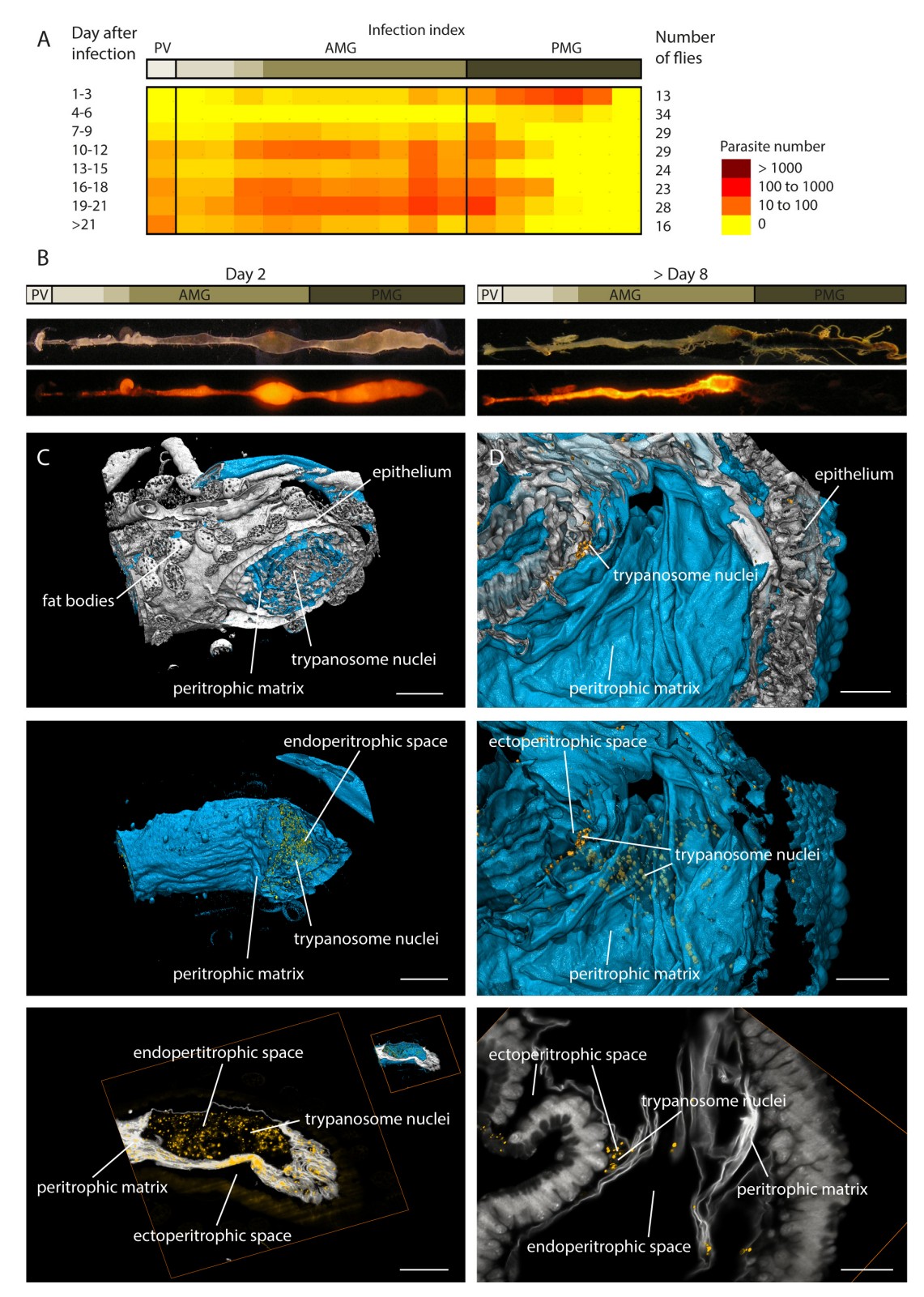

**Figure 3.** Trypanosome midgut infections can be monitored dynamically and with single parasite resolution. (**A**) Heat map of the early infection process, analysed by the amount of fluorescent trypanosomes in different areas: proventriculus (PV), anterior midgut (AMG) and posterior midgut (PMG). Trypanosomes are located in the posterior midgut between 1–3 days after the infective blood meal. The flies´ immune system clears a large part of the parasite population between days 4–6 (*Aksoy et al., 2003*; *Van Den Abbeele et al., 1999*). In flies that were not able to eliminate all parasites,

*Figure 3 continued on next page*

*Figure 3 continued*

the population expands again, while the trypanosomes concentrate in anterior regions. The midgut stays infected for the remaining life-span (*Gibson and Bailey, 2003*). (B) The distribution of trypanosomes at day two after the infective meal is shown on the left. The typical infection pattern after a stable infection (>8 days) is shown on the right, where the trypanosomes have invaded the ectoperitrophic space. (C) and (D) show surface rendering models of isolated infected fly guts. The intestinal tissue is visualised by autofluorescence (grey). The PM is stained with rhodamine-labelled WGA (cyan) and the trypanosome nucleus with a GFP-reporter (yellow). (C) Dissected part of the midgut 2 days post-infection. The PM is shown isolated in the middle panel, together with the fluorescent trypanosome nuclei, which are located exclusively inside the internal midgut lumen (animated in *Video 3*). In the bottom panel a single plane shows the localisation of the trypanosomes within the endoperitrophic space (animation of full stack in *Video 3*). Scale bars: 100 µm. (D) Dissected midgut >day 8 post-infection. The top view is onto the inside surface of the PM (view point in the gut lumen), with underlying epithelial tissue and trypanosomes between folds of the PM belonging to the ectoperitrophic space. The middle panel allows the same view, albeit with the PM rendered transparent, in order to visualise the trypanosomes concentrated around the outer surface of the PM. The single slice in the bottom panel resolves groups of nuclei in the ectoperitrophic space whereas the endoperitrophic space is void (animation of full stack in *Video 3*). Scale bars: 50 µm. *Figure 3—Videos 1* and *2*. Infective metacyclic cells expressing nuclear GFP (*Figure 3—Video 1*) or PFR-GFP (*Figure 3—Video 2*) are imaged immediately after release from the salivary glands, showing that the transgenic cell lines successfully complete the developmental cycle in the tsetse fly. *Figure 3—Video 3*. Multicolour light sheet fluorescence microscopy details the peritrophic matrix within the tsetse midgut and allows identification of trypanosomes with single cell accuracy (*Figure 3—Video 3*).

**Figure 3—Video 1.** Metacyclic cells expressing nuclear GFP successfully complete the developmental cycle in the tsetse fly.

**Figure 3—Video 2.** Metacyclic cells expressing PFR-GFP successfully complete the developmental cycle in the tsetse fly.

**Figure 3—Video 3.** Multicolour light sheet fluorescence microscopy details the peritrophic matrix within the tsetse midgut and allows identification of trypanosomes with single cell accuracy.

allowed the flagellum to be traced. The data was further used to generate surface-models, which allowed the course of the traced flagellum along and around the body surface to be visualised (*Figure 5*).

For the cellular waveform analysis, we recorded hundreds of freshly released trypanosomes. From these, persistently forward swimming trypanosomes were selected and used to trace the cell shapes in consecutive frames (*Video 6*). The resulting shapes were plotted for the duration of one flagellar beat in 3D, using time as the z-axis (*Figure 6*, *Video 6*, white). The 3D-visualisations show the oscillation of the flagellar tip (*Figure 6*, blue). The travelling waves are apparent and information on amplitude and wave length of each progressive flagellar wave is quantifiable as it travels to the posterior end.

Eight different developmental (transition) stages were analysed, excluding only epimastigote trypanosomes attached to the epithelium of the salivary glands and the long epimastigote stage that presumably degenerates after asymmetric division. The fly vector microswimmers (*Figure 5A–G*) were compared to the two main bloodstream forms (BSF) (*Figure 5H,I*) (*Bargul et al., 2016*).

None of the life cycle stages revealed an undulating membrane, which corroborates the finding that propulsion of bloodstream form trypanosomes is solely accomplished by flagellar beating according to resistive force theory, because the membranes of flagellum and cell body are directly adjacent and do not form an extended, fin-like appendage (*Heddergott et al., 2012*). The procyclic trypanosomes (PCF) (*Figure 5A*) develop in the gut lumen from the cell cycle arrested stumpy BSF (*Figure 5I*). Compared to the bloodstream forms, the procyclic cells have elongated and thinned considerably. The posterior end is more pointed and the distance of the flagellar pocket from this end has increased by several micrometres. The

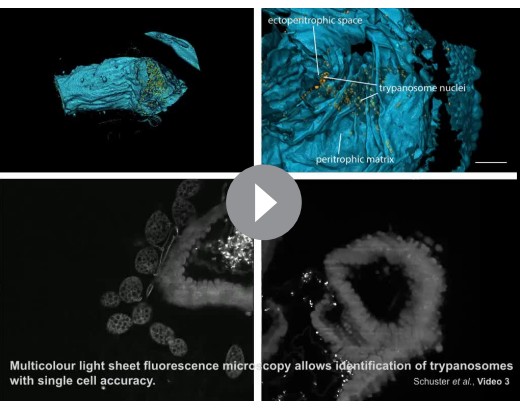

**Video 3.** Multicolour light sheet fluorescence microscopy allows identification of trypanosomes with single cell accuracy.

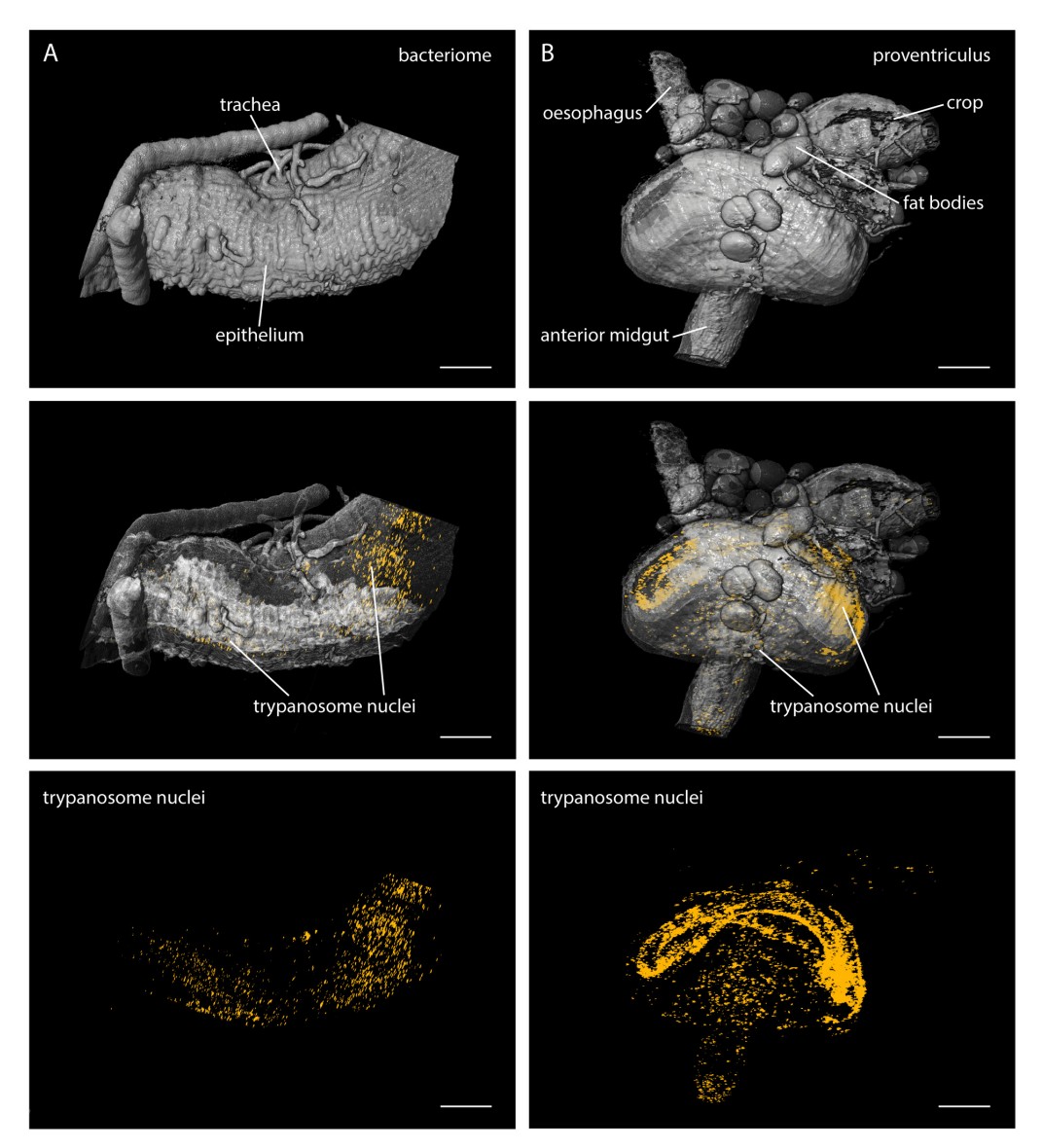

**Figure 4.** LSFM quantifies trypanosome populations in complex tsetse organs. Infected tsetse organs were surgically removed and rendered using LSFM autofluorescence data in the top panel (grey). Nuclear GFP reveals the distribution of trypanosomes in the corresponding volume in the lower panels (yellow). The middle panel shows the merged 3D-localisations with semi-transparent organ models. (**A**) The bacteriome is located in the anterior midgut region and harbours endosymbiotic bacteria. It has a characteristic three-dimensional structure which is discernible with the appropriate transparency settings, due to higher autofluorescence levels (middle panel). The trypanosomes are excluded from this organ and located in the ectoperitrophoic space around the juxtaposed PM (not labelled in this specimen). (**B**) The proventriculus connects the anterior midgut, the crop and the salivary duct. The PM (not labelled) is produced here by a ring of specialised cells. Trypanosomes accumulate in partially high cell densities around this toroid structure. Scale bars: 100 µm. *Video 4* contains animations of the proventriculus model and of the corresponding LSFM data stack.

flagellum describes a 180° right hand turn around the cell body progressing towards the anterior end. This is similar to the flagellar course in BSF, but as the flagellar pocket is located further anterior, the flagellum wraps around the cell body at a more anterior position. These characteristics of PCF morphology result in the seemingly 'flipping' forward swimming movement observed in high speed videos, as the relatively stiff posterior end is not deformed by the flagellar wave. The rotation frequency of procyclic parasites is similar to that of BSF cells, albeit the rotation is easier to observe in the latter as the flagellar wave travels to the posterior end and deforms the whole length of the

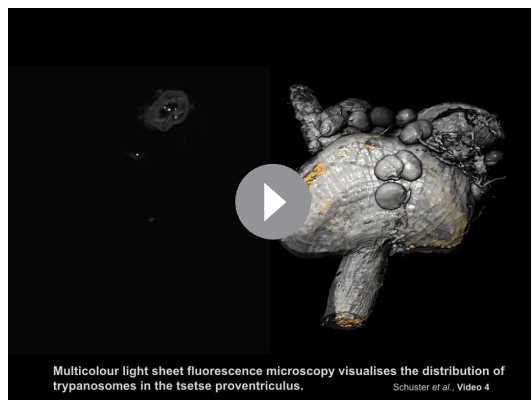

**Video 4.** Multicolour light sheet fluorescence microscopy visualises the distribution of trypanosomes in the tsetse proventriculus.

elastic cell body, giving BSF swimmers a smoother appearing auger-like trajectory, which was eponymous for the parasite.

The 'long procyclic' cells shown in *Figure 5B* represent a transitional state, whose morphology approaches the mesocyclic type (*Figure 5C*). Note that morphology does not discriminate between early and late procyclic forms expressing different surface proteins (*Knüsel and Roditi, 2013*). For the purpose of this work, we have explicitly analysed the morphologically distinct transition stages and their characteristic behaviour as microswimmers in their distinctive environment. The long procyclic trypanosomes isolated from the flies had elongated and thinned further, without changing the attachment conformation of the flagellum.

Both procyclic cell types revealed similar cellular waveforms (*Figure 6A,B*). The long procyclic trypanosomes maintain a slightly higher frequency and amplitude of flagellar oscillation. Additionally, the traveling flagellar wave seems not to be dampened by the posterior end of the cell body as strongly as in the shorter cell. This could be the result of the slightly thinner and more flexible posterior cell end. Together with the increased absolute length of the flagellum, these parameters allow the long procyclic to reach more than double the speed of the shorter form.

The development of cell cycle-arrested mesocyclic cells is characterised by the disappearance of the free flagellar part at the anterior end, due to further elongation of the cell body (*Figure 6C*). The nucleus had moved further away from the kinetoplast to keep its position roughly in the cell centre. Altogether the cells had a flattened and straight appearance, which coincides with the 180° flagellar turn around the cell being stretched over a longer distance, that is the flagellum spirals around the cell body with a smaller helix angle. Due to the thicker anterior cytoplasmic part, the flagellum often seemed to be able to turn further around the cell body, almost completing a 360° turn.

The transition to the cellular waveform of mesocyclic cells is characterised by a switch to traveling waves of smaller amplitude and wavelength (*Figure 6C*). Despite constant beat frequency and further increased flagellar length, swimming speed is slightly reduced. The cellular waveform coincides with the anteriorly extended cell body (*Figure 5C*). The lack of a free anterior part of the flagellum is most likely responsible for the characteristic change of waveform and results in very straight swimming trajectories.

The transformation of mesocyclic cells into epimastigote forms occurs by a continuous repositioning of the cell nucleus from the anterior to the posterior side of the kinetoplast, followed by re-entry into the cell cycle (*Hoare and Wallace, 1966*; *Sharma et al., 2008*). The cytoplasm is also continuously reduced, finally leaving the flagellum all but free and the epimastigote microswimmer with an almost sperm-like appearance (*Figure 5D–E*). During the transition from mesocyclic to epimastigote parasites, a continuously larger part of the anterior flagellum protrudes, which results in higher amplitude waves being initiated by the free flagellar tip (*Figure 6D*). While the flagellar length only increases slightly, the beat frequency doubles, allowing the transition stages to reach speeds above 100 µm/s.

The epimastigotes possess a virtually completely free flagellum (*Figure 5E*), producing large amplitude waves that travel unhindered to the posterior cell body, enabling to be pulled at speeds well above 100 µm/s (*Figure 6E*). Cell division is rapidly initiated in epimastigote trypanosomes with the kinetoplast dividing first. The epimastigote cell divides asymmetrically to produce a long and a short epimastigote (*Figure 5F*). The short epimastigote cells are characterised by a rudimentary flagellum attached anteriorly to a thin and stiff cell body (*Figures 5F* and *6F*). The flagellum is capable of regular oscillation and persistent movement, but the cell does not reach speeds above 10 µm/s, as there is no appreciable travelling wave propagated. Persistent movement is also not sustained in

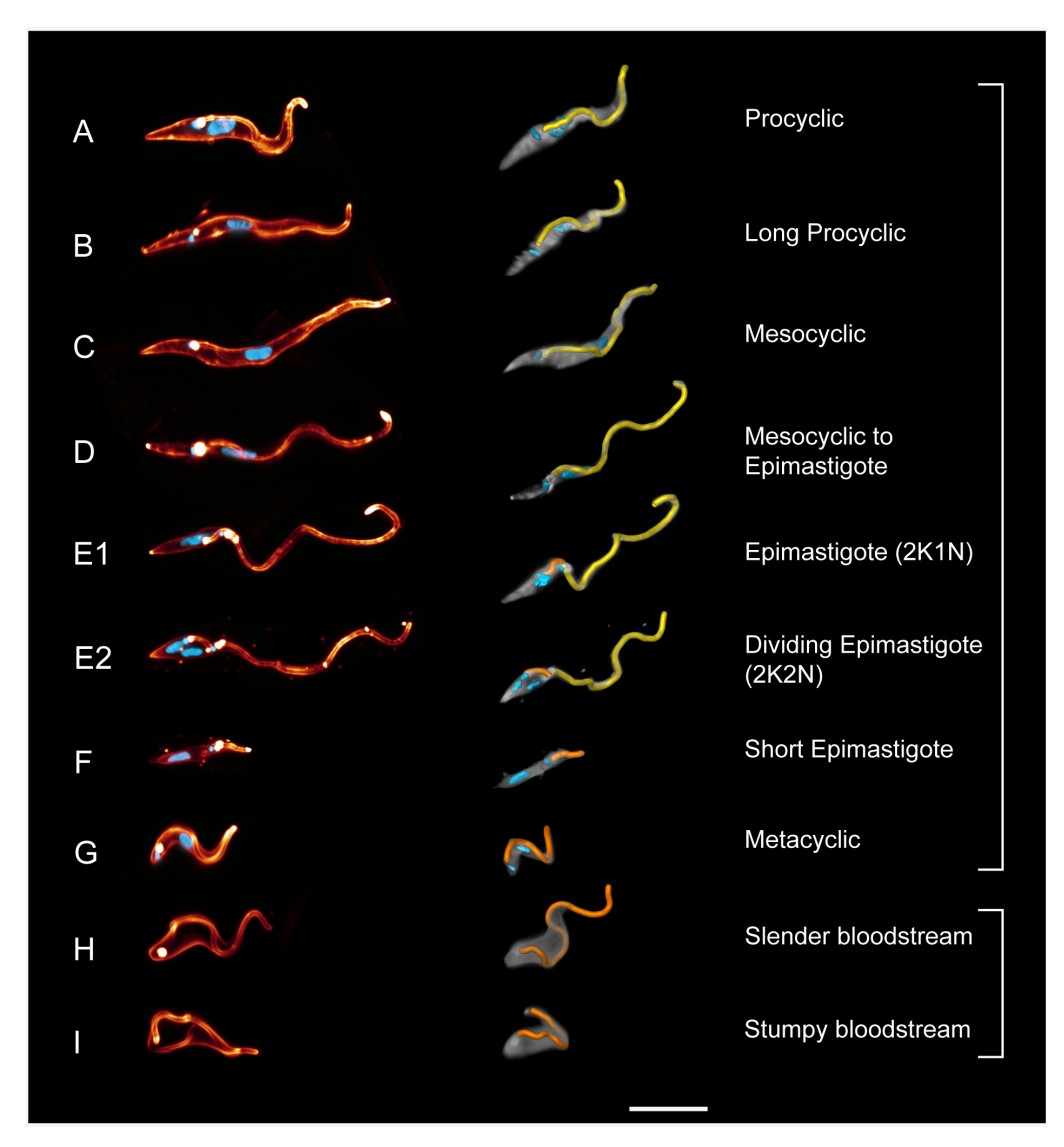

**Figure 5.** High-resolution 3D-morphometry of all trypanosome life cycle stages. Trypanosomes were isolated from infected tsetse flies (**A–G**) or mice (**H–I**). The cell surface was fluorescently labelled with a sulfo-NHS dye (red) and the cell nuclei and kinetoplasts were labelled with DAPI (cyan). In the left panel, representative 3D-volume models of surface-labelled parasites are shown. The right panel presents the corresponding 3D surface models, with the cell body in grey and the attached flagellum in yellow or orange. Scale bar: 10 µm. Procyclic cells (**A**) exhibit a 180° right hand turn of the flagellum around the cell body. Long procyclic cells (**B**) are larger than normal procyclics, but reveal the same characteristic flagellar attachment. Mesocyclic cells (**C**) show a more elongated cell body and a straighter flagellar attachment, also fulfilling a 180° flagellar turn, but lack a free flagellar tip. During the transformation from mesocyclic to epimastigote cells (**D**), the nucleus elongates and moves posterior, and the cytoplasm at the anterior tip retracts. After repositioning, the epimastigote cell begins cell division, with the kinetoplast duplicating first (E1, 2K1N configuration). The orange flagellum represents the new flagellum of the adolescent short epimastigote daughter cell. Shortly before cytokinesis, the dividing epimastigote cell (E2, 2K2N configuration) shows an almost sperm-like appearance. The short epimastigote cell (**F**) resulting from this division has a thin, straight cell body with an extremely reduced flagellum protruding from the anterior end. Infective metacyclic cells (**G**) in contrast, have a curly appearance with a 180° turn of the flagellum, which originates from the posterior end. Metacyclic cells thus re-establish the trypomastigote configuration, which is maintained in the slender (**H**) and stumpy bloodstream forms (**I**). (H, I, adapted from [***Bargul et al., 2016***]). (***Video 5*** contains animations of the 3D-surface models)

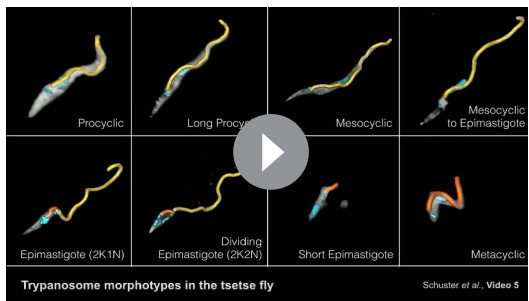

**Video 5.** Trypanosome morphotypes in the tsetse fly.

most cells. Thus, within one cell cycle, a very capable microswimmer produces an almost immotile parasite.

Metacyclic parasites reveal trypomastigote morphology, as the nucleus has repositioned to the anterior side of the kinetoplast (*Figure 5G*). They swim freely in the saliva and represent the infective form that is pre-adapted to survive in the mammalian host (*Otieno and Darji, 1979*) (*Figure 5H,I*). The cell body appears elastic and is deformed in the shape of the flagellar waveform. The flagellum emerges at the very posterior end and has once more adopted the right hand 180° turn around the cell body. The final insect stage, once more represents a very different type of microswimmer. The effect of the changed morphology (*Figure 5G*) on swimming behaviour is immediately obvious in the visualisation of the metacyclic cellular waveform (*Figure 6G*). The elastic cell body deforms and oscillates with the attached flagellum. The flagellar tip produces waves of high amplitude with frequencies similar to procyclic parasites. A travelling wave propagates along the cell body, but the steep chiral course of the attached flagellum produces relatively high rotational forces, which result in a contracted cellular waveform, reducing forward swimming speed. Following transmission to the mammalian host, the metacylic stage differentiates to the slender bloodstream stage. The cycle begins anew when the slender population produces the fly infective stumpy bloodstream form (*Figure 5I*) (*Vickerman, 1985*).

The measurement of the remarkable dynamic pleomorphism of tsetse trypanosomes completes the second aim of our endeavour, namely to unravel the 3D cell architecture and motion performance of all relevant trypanosome microswimmer types.

## Trypanosome motion in the fly is characterised by successions of solitary and collective motion

Now we had gained an overview of (1) the topological constraints of the tsetse habitat, (2) the occurrence of trypanosome infections therein, and (3) the morphodynamic properties of the individual parasites types, we proceeded to analyse the parasites' swimming behaviour within the tsetse fly. In order to elucidate the motility on the single cell level, we performed live high-speed analyses in defined regions of the alimentary tract. For this purpose, we dissected the respective tissues, in order to visualise the parasites by high resolution DIC microscopy.

During their journey through the various organs and tissues of the tsetse fly trypanosomes pass developmental 'bottlenecks' (*Dyer et al., 2013*; *Oberle et al., 2010*). In the procession of these events, a certain number of parasites move into new environments, where they must be able to proliferate to larger population sizes. Thus, the tsetse system should allow to comparatively analyse the individual morphotypes, both as solitary swimmers (*Figure 7*), or in groups of various sizes and densities (*Figure 8*), in the tissues or interstitia of different fly organs.

**Video 6.** Visualisations of the dynamic cellular waveforms of different trypanosome morphotypes.

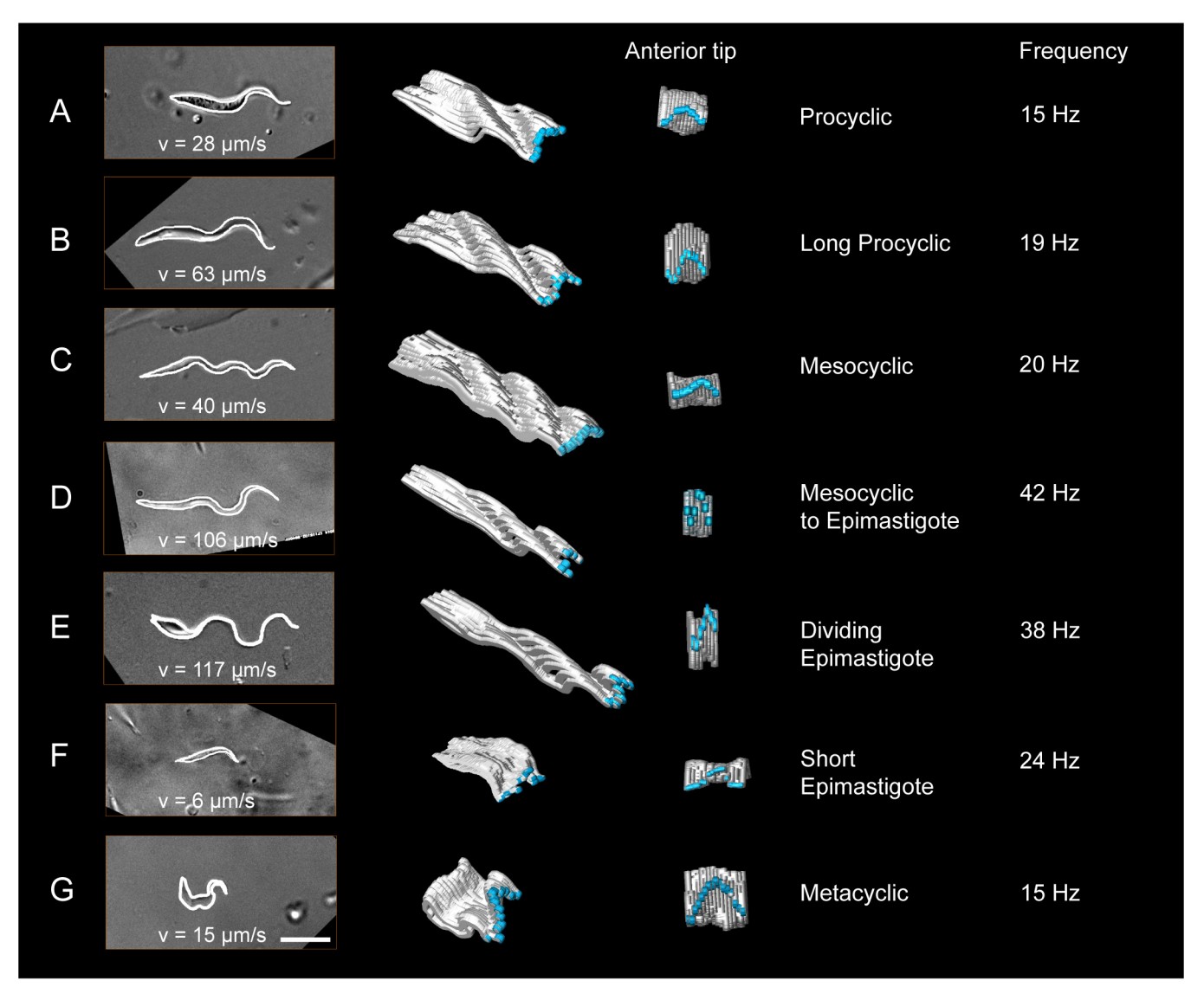

**Figure 6.** Measurement of the dynamic cellular waveforms of different trypanosome morphotypes. Images in the left panel are stills from videos, captured with a frame rate of 250 fps. The speed (v in μm/s) represent the highest velocity reached during the video sequence analysed. The panel in the middle is a model of the outline for one single flagellar beat analysed frame by frame. The frames were stacked along the time axis in a three-dimensional surface representation, which allows the visualisation of the cellular waveform produced by the flagellum and the cell body in two different views. The flagellar tip was highlighted in blue. The model shows the travelling waves running along the cell body in a top-diagonal view and one wavelength of the flagellar beat in the view of the anterior tip. The frequency (Hz) of the analysed flagellar beat is shown on the right. Procyclic cells (**A**) and long procyclic cells (**B**) show similar waveform patterns, although the long procyclic cells generally swim faster. Mesocyclic cells (**C**) show a characteristic waveform due to their small amplitude during flagellar beating. When they start differentiating from mesocyclic to epimastigote cells (**D**), the amplitude increases again with a higher frequency and cells gain more speed. Dividing epimastigote cells (2K2N) (**E**) have proven to be the fastest swimmers of tsetse fly stages. Short epimastigote cells (**F**) are weak swimmers, despite beat frequencies similar to procyclic cells, due to their lack of a free flagellum. Infective metacyclic (**G**) cells show an increase in amplitude and a characteristic curly waveform, while reaching medium beat frequencies and swimming speeds. *Video 6* contains all original video sequences selected for waveform analysis. The videos of trypanosome stages are consecutively played to show the position of each traced waveform along the time-axis in the 3D-models.

## Solitary microswimmers in the tsetse fly

After their uptake by the tsetse fly, the trypanosomes swim freely between blood cells in the gut lumen (*Figure 7A*, *Video 7*). Here, the BSF, ingested with the mammalian host blood, transform into

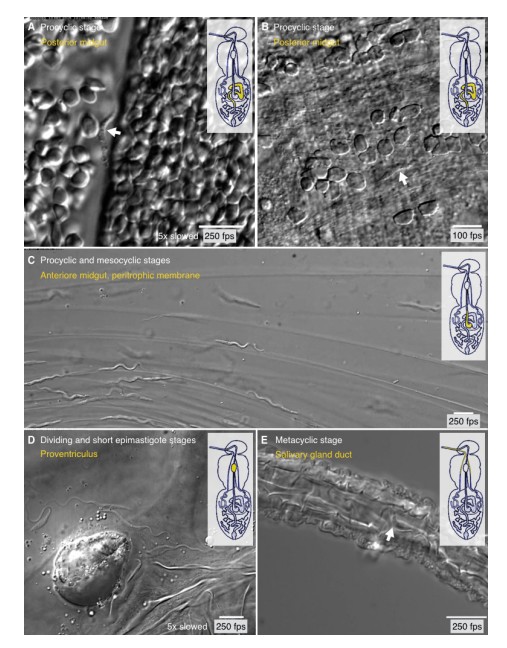

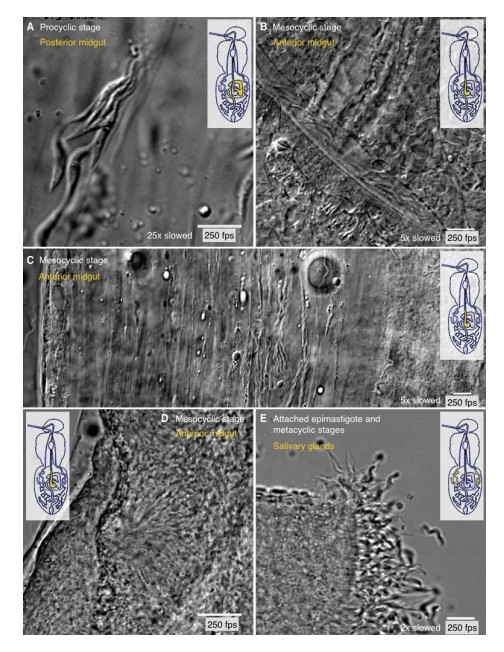

**Figure 7.** Trypanosome life cycle stages as solitary microswimmers in varying tsetse environments. Infected tsetse flies were dissected in PBS and various regions, marked in the inset cartoon fly, were analysed by high speed microscopy (100 to 250 fps). Images are stills of the corresponding videos (*Video 7*), showing trypanosomes (white arrows) of various developmental stages. Scale bars: 10 µm. (**A**) Procyclic trypanosome swimming between blood cells in the posterior midgut lumen shortly after feeding. (**B**) A procyclic cell in tissue of the posterior midgut, confined by gut epithelium. (**C**) Procyclic to mesocyclic transition stages swimming along sheets of dissected PM in the anterior midgut. Trypanosomes experience different degrees of confinement and display characteristic straight trajectories and U-turns. (**D**) Various epimastigote cells inside the proventriculus, confined to a limited fluid-filled cavern. (**E**) Single metacyclic cells inside the thin salivary gland duct. The cells are motile, but mainly tumble around one position, as they await the tsetse fly´s next blood meal. *Video 7* plays the original videos simultaneously with the annotated speeds.

**Figure 8.** Different degrees of trypanosome crowding and environmental confinement can be found throughout the fly. Infected tsetse flies were dissected in PBS and various regions, marked in the inset cartoon fly, were analysed by high speed microscopy (250 fps). Images are stills of the corresponding videos (*Video 8*). Scale bars: 10 µm. (**A**) Procyclic cells at day two after infection in the posterior midgut show the ability to form clusters and synchronise their flagellar oscillations. (**B**) Long procyclic to mesocyclic transition stage cells packed within a channel in the anterior midgut tissue of a late stage infected tsetse. (**C**) Mesocyclic cells in anterior midgut tissues and encased in folds of the PM. Depending on the degree of confinement, partly synchronised clusters of cells are visible. Strongly confined single cells display significant bending of the cell body and are able to perform sharp U-turns in the limited space. (**D**) High density swarms of mesocyclic cells inside the midgut ectoperitrophic space create superordinate wave patterns and generate tissue deforming force. (**E**) Sliced salivary gland with epithelium-attached epimastigote cells and free pre-metacyclic cells floating in the surrounding medium. The intact tissue was too dense to allow imaging of salivary gland stages by light microscopy, therefore the organ was dissected to show the free posterior ends of attached epimastigote trypanosomes in a limited region. *Video 8* plays the original videos simultaneously with the annotated speeds.

PCF. Erythrocytes are densely packed in the gut, but there are also less dense regions where individual PCF cells can be traced. The cell marked in *Figure 7A*, for example, represents a typical procyclic trypanosome, which swam persistently with an average speed of 69 µm/s and a flagellar beat frequency of 25 Hz (*Video 7*). In *Figure 7B* a comparable procyclic cell is shown confined in a space between gut epithelium and blood cells. It swam slowly with reduced amplitude, was repeatedly halted, and reversed the flagellar beat in order to change direction and move forward along a new path (*Video 7*). This behaviour is characteristic

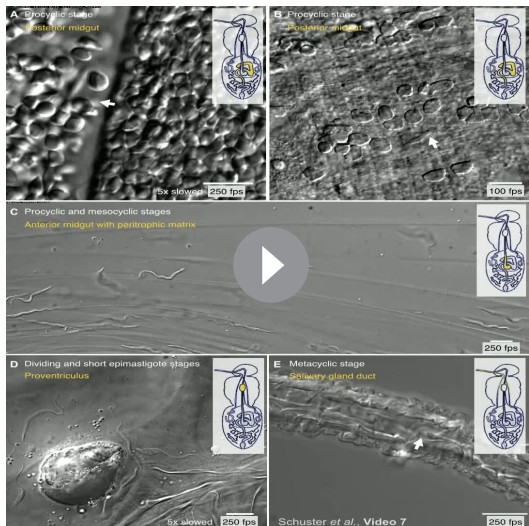

**Video 7.** Trypanosome life cycle stages as solitary microswimmers in varying tsetse environments.

also for BSF trypanosomes that experience mechanical hindrance, and is postulated to allow the parasites to escape from dead ends, for example in confined tissue spaces (*Heddergott et al., 2012*).

After reaching the ectoperitrophic space, the procyclic cells continue development by transforming into the cell cycle arrested mesocyclic stage. Transition stages and mesocyclic forms can be frequently seen swimming near the walls formed by the PM in *Figure 7C* (*Video 7*). This video also shows the cells experiencing various degrees of confinement. Several cells swam along the matrix with the characteristic waveforms of long PCF or mesocyclic trypanosomes, others were stuck in narrow folds of the PM and showed a reduction in wave length and amplitude, before they reversed waves and backed out of their trap. Mesocyclic parasites are also capable of sharp U-turns, due to their highly flexible cell body (*Figure 8C*, *Video 8*).

Some mesocyclic cells eventually navigate through the ectoperitrophic space to the proventriculus, where they undergo a continuous transition to the thin epimastigote cells, which are propelled by a sperm-like, long free flagellum. *Figure 7D* presents two mesocyclic to epimastigote transition forms, displaying short sequences of forward movement in the corresponding video, but the flagellar waves stay ineffective, the cells reverse direction frequently and do not leave the observed area in the proventriculus (*Video 7D*). As shown in the waveform analyses above (*Figure 6D*), outside of such enclosed cavities, the epimastigote forms can reach high directional swimming velocities.

The epimastigote cells undergo an asymmetrical division to produce short epimastigotes that presumably go on to infest the salivary glands. Dividing and short epimastigote cells swimming in the interstitial area are exemplified in *Video 7D*. The sharp drop in swimming capability in the short product of the asymmetric division is obvious. For this reason, the dividing epimastigote stage is believed to be a piggyback transporter for the developmentally relevant short epimastigote cells, although this route to the salivary glands has not been demonstrated so far (*Sharma et al., 2008*; *Van Den Abbeele et al., 1999*).

Finally, *Figure 7E* (*Video 7E*) shows two representative metacyclic cells in the salivary duct. The cells tumbled, meaning they frequently reversed flagellar beats and effectively stayed in one place. This could be expected to be normal in vivo behaviour for the infective metacyclic form, as it lingers in the salivary glands for the next blood meal of the tsetse fly, which will inject the parasite into a vertebrate host.

The above data illustrate that the solitary swimming behaviour of all trypanosome development stages is massively influenced by the microenvironment, especially by way of confinement and presence of boundary layers. Near-wall motion is a prominent feature of mesocyclic

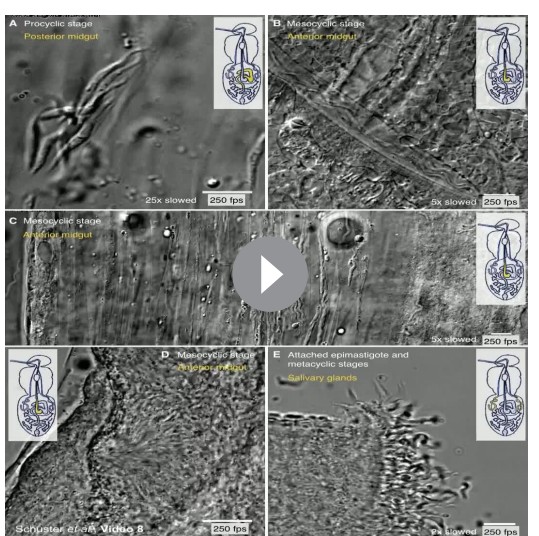

**Video 8.** Different degrees of trypanosome crowding and environmental confinement can be found throughout the fly.

microswimmers, which not only is biologically relevant, but also provides theoretical physics with an example of a natural near-wall swimmer.

## Collective microswimmers in the tsetse fly

While tsetse trypanosomes can be observed as solitary swimmers, they also show highly coordinated collective motion behaviours, over a large range of cell concentrations and in a multitude of variably confining microenvironments (*Figure 8*, *Video 8*).

Most trypanosome life cycle stages were observed to collect in various physical niches where no further forward movement seemed possible. As these cells usually belonged to one and the same developmental stage, all neighbours exhibited very similar characteristic flagellar oscillations. Interestingly, those groups of cells frequently synchronised their flagellar beats. *Figure 8A* exemplifies a group of procyclic parasites. Cells swam forward to join the group and synchronised, but also reversed their flagellar beat, which resulted in backward motion to leave the cluster (*Video 8*). This behaviour is not stage-specific, as *Figure 8B* (*Video 8*) shows a similar situation with mesocyclic cells. Several groups of mesocyclic cells can be trapped in close proximity within neighbouring PM folds and partly synchronise their oscillations (*Figure 8C*, *Video 8*).

At very high densities, especially mesocyclic cells were seen to fill large interstitial spaces. The cells were packed closely, continuously oscillating and obscuring most details of the surrounding tissue (*Figure 8D*, *Video 8*). Several regions were visible where the tips of trypanosomes seemed to converge and began to synchronise their flagellar waves. Large numbers of trypanosomes gathered around these centres and produced superordinate wave patterns. This collective motion was even seen to move a part of the tissue, thus apparently producing significant directional force.

Finally, we compared the motility of the attached epimastigote parasites in the salivary gland, where cells again proliferate to high densities (*Figure 8E*, *Video 8*) (*Rotureau et al., 2012*; *Vickerman, 1985*). In contrast to the cells in mesocyclic collectives, the epimastigote cells attach with their anterior ends to the gland epithelium and thus are not free to self-organise. Single pre-metacyclic cells, produced by another asymmetric division (*Rotureau et al., 2012*), are continuously released into the saliva where they develop into metacyclic forms. Vigorous forward and reverse flagellar beating could be observed, but it was not possible to analyse the organisation status within the intact organ (*Video 8*).

## Motile behaviour of single trypanosomes in tissues and clusters

Having documented the principal capabilities of tsetse trypanosomes for collective behaviour, we next extracted the motion pattern of single cells within large swarms and in dense tissues. For this, we generated transgenic parasites expressing GFP-fusion proteins within the flagellum, the nucleus or both. Tsetse flies were infected and all developmental stages could be fluorescently traced. The flagellar marker even enables us to measure the oscillatory parameters on the single cell level. The nuclear marker was especially useful, as the bright signal facilitated cell tracking. Furthermore, the observation of the characteristic changes in nuclear shape allowed the identification of different developmental stages in mixed clusters: Procyclic parasites have a round nucleus, whereas mesocyclic trypanosomes feature an elongated nucleus. During the mesocyclic to epimastigote transition, the nucleus elongates further to assume a more rod-shaped appearance (*Sharma et al., 2008*).

High-speed microscopy of fluorescent reporter trypanosomes thus not only allowed the quantitative analysis of single cell motion within swarms, but also the identification of the life cycle stage of the individual parasites (*Figure 9*, *Video 9*).

As an example, *Figure 9A* shows a typical population of predominantly mesocyclic cells that was traced in the ectoperitrophic space of the anterior midgut. The anterior leading edges of the partly synchronised clusters pushed against an epithelial border (*Figure 9A1*, *Video 9*). Only few trajectories of the fluorescent nuclei showed persistent swimming, indicating that most tracked cell nuclei belonged to clustering cells (*Figure 9A3*). In comparison, a region in the proventriculus without synchronisation patterns, containing predominantly mesocyclic to epimastigote transition stages, exhibited a far greater proportion of persistently swimming cells, probably due to the lower density of neighbouring cells and absence of a continuous border (*Figure 9B3*). Additionally, a bias in swimming directions was observed in the selected region. This effect could be due to topographical

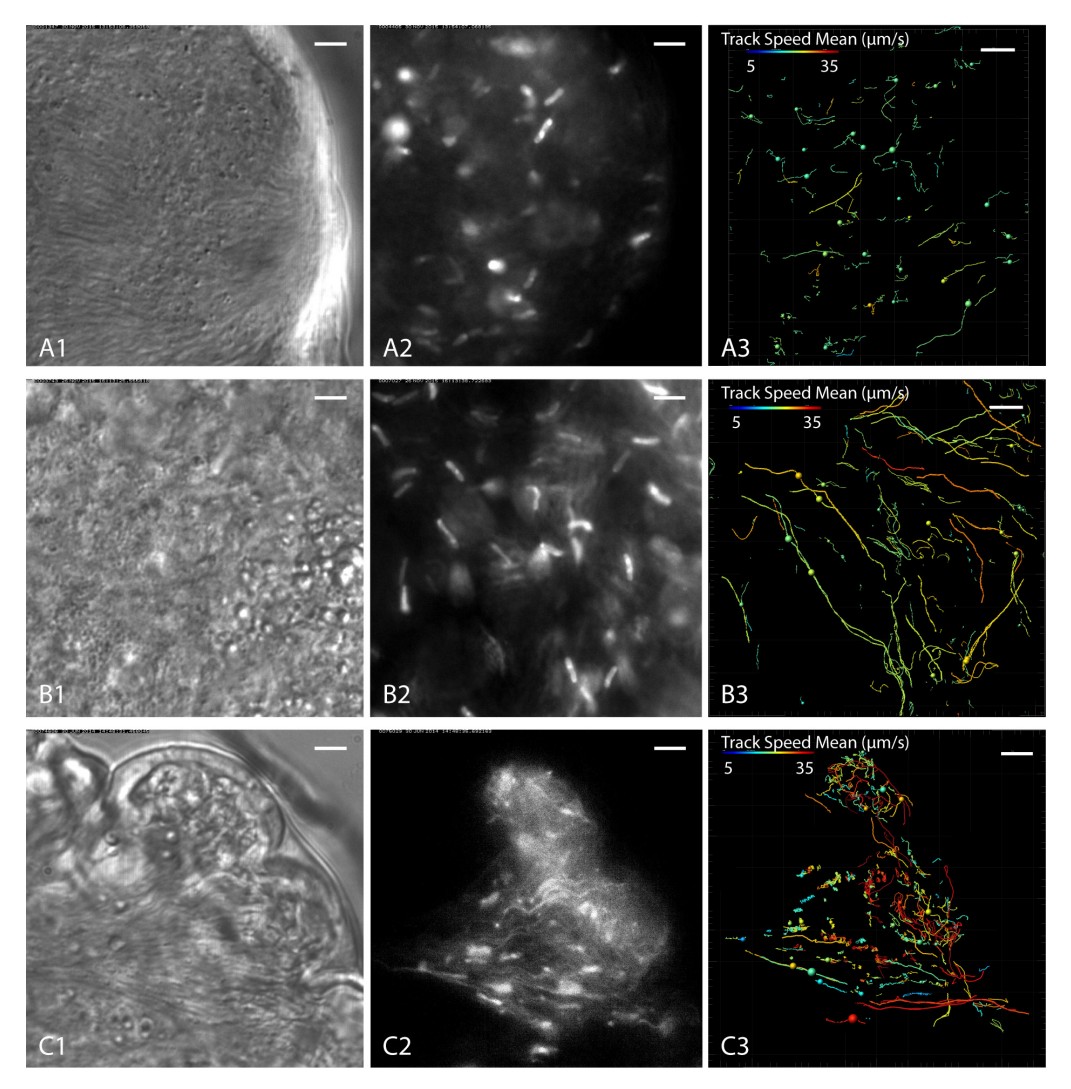

**Figure 9.** Cell tracking details single trypanosome motion behaviour in parasite clusters. Infected tsetse flies were dissected in PBS and specific areas containing high trypanosome concentrations selected. Fluorescent trypanosomes express GFP signal in the nucleus (**A and B**) or additionally in the flagellum (**C**). The images are stills of the corresponding videos captured with 250 fps (*Video 9*). The left panel shows transmitted light images, the middle panel shows fluorescence images of the same region and the right panel the corresponding tracking analysis. The mean track speed is colour coded and shows a range of 5–35 µm/s. Scale bars: 5 µm. (**A**) Accumulation of mesocyclic trypanosomes inside the ectoperitrophic space in anterior midgut tissue. The trypanosomes show less persistent swimming in this area, due to dense synchronised clusters of parasites. (**B**) Mesocyclic cells and mesocyclic to epimastigote transition stages with elongated cell nuclei inside the proventriculus. There are more persistent swimmers in this region, probably due to lesser cell crowding and topographical structures effectively producing microswimmer channels. (**C**) Trypanosomes labelled with a nuclear and/or a flagellar marker, in anterior midgut tissue, experiencing different levels of confinement in close proximity. In the left region a high degree of clustering and synchronisation is obvious, whereas to the right, fast single parasites are tracked swimming in fluid-filled cavernous regions. Single parasites are tracked swimming into the cluster at the left, synchronise their oscillations temporarily and eventually reverse swimming direction and leave the swarm. *Video 9* shows the original videos and the synchronous animated tracking data in original speed.

constraints of the tissue, which might define proper ´channels´, guiding persistent microswimmers (*Figure 9B1–3*).

In a region of the midgut containing a synchronising cluster as well as fluid-filled free spaces (*Figure 9C1*), the traces show little persistence of 'trapped' swimmers, whereas parasites outside of clusters moved persistently, being only confined by the epithelial borders (*Figure 9C3*). Notably, single cells could be traced swimming into the flock, where they synchronise with the stationary

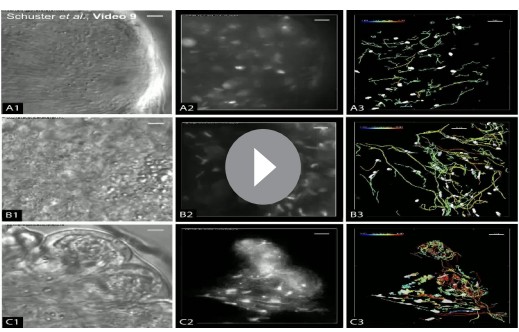

**Video 9.** Cell tracking details single trypanosome motion behaviour in parasite clusters.

oscillatory movements of the surrounding trypanosomes, eventually reverse the flagellar beat and leave again (*Video 9C2*). This indicates the self-organisation of the assemblies by hydrodynamic forces.

The average speeds of persistently swimming cells in these tissues were measured to be around 15–25 µm/s (colour coded in *Figure 9A3–C3*), with the highest speeds of around 40 µm/s measured in fluid filled spaces (red tracks in *Figure 9C3*).

High-speed fluorescence microscopy and automated cell tracking allowed us to complete the third part of our endeavour, namely the quantification of both solitary and collective trypanosome motion in diverse tsetse fly environments. We were able to identify and track single trypanosomes in tissues, swarms and during their transition between free swimming and confined oscillation.

## Self-organisation of parasites by hydrodynamic interaction

The experiments so far demonstrated that the journey of trypanosomes through the tsetse fly is marked by successions of solitary and collective motion patterns. The distinct life cycle stages are differentially equipped for this endeavour, which is reflected in their cellular waveforms. The impressive synchronisation of large groups of densely packed, oscillating cells raised the question if this behaviour was driven by quorum sensing, that is by chemical cues, or by hydrodynamic self-organisation processes. Procyclic trypanosomes have been shown to perform social motility in vitro, albeit in a manner probably controlled by chemosensing and over long time scales (*Imhof et al., 2014*; *Oberholzer et al., 2010*, *2015*). A chemical process would be reaction-diffusion driven, and hence, should generally be slower than sensing by hydrodynamic interactions. Thus, we attempted to measure the onset of cell synchronisation. In fact, we observed synchronised motion patterns to form and disperse spontaneously (*Figure 10*, *Video 10*).

In the ectoperitrophic space of the anterior midgut, we observed several regions densely populated with cells in or close to the mesocyclic developmental stage. These were often surrounded by epithelial tissues forming cavernous structures where the cells were observed to swim freely in all directions (*Figure 10A* left, *Video 10* compare similar structures in *Figure 9C1*). In one of these regions we recorded an alignment of cells several seconds later, where all cells in the region were seen beating in one direction, apparently constricted at the narrow corner of the epithelial border. The cells had arranged in a swarm conformation that was stable for at least several minutes (*Figure 10A* right, *Video 10*). The trigger for this spontaneous synchronisation event was unlikely of chemical nature, but rather a shift in the surroundings of the dissected specimen, which could have changed the volume of, or the pressure on the parasite-harbouring space.

In another recording of synchronised cells with fluorescent flagellar markers (*Figure 10B* left, *Video 10*), the reversal of such a swarm into an isotropic distribution of swimming cells was observed, also on a timescale of seconds (*Figure 10B* right, *Video 10*). The triggering event for this change is also unknown, but, in reversal of the presumable confining event in *Video 10A*, it could be speculated to be a release of space-confining pressure of the surrounding tissue in *Video 10B*.

These results showed unambiguously that the collective behaviour of large trypanosome populations can be self-organised by hydrodynamic interactions without the need for any physical attachment to tissues, as had been shown for small groups of cells (e.g. *Figure 8A–B*). It should be noted, that the timescale of a few seconds observed here for collective organisational status change is significantly smaller than the shortest known timescale of potential chemotactic behaviour in trypanosomes (*Oberholzer et al., 2010*). Although chemical signals will undoubtedly be relevant for rapid fluctuations of flagellar beating, an adaptive reaction system required for directed behaviour of parasite swarms is unlikely to be responsible for establishing the switching behaviour observed in our experiments.

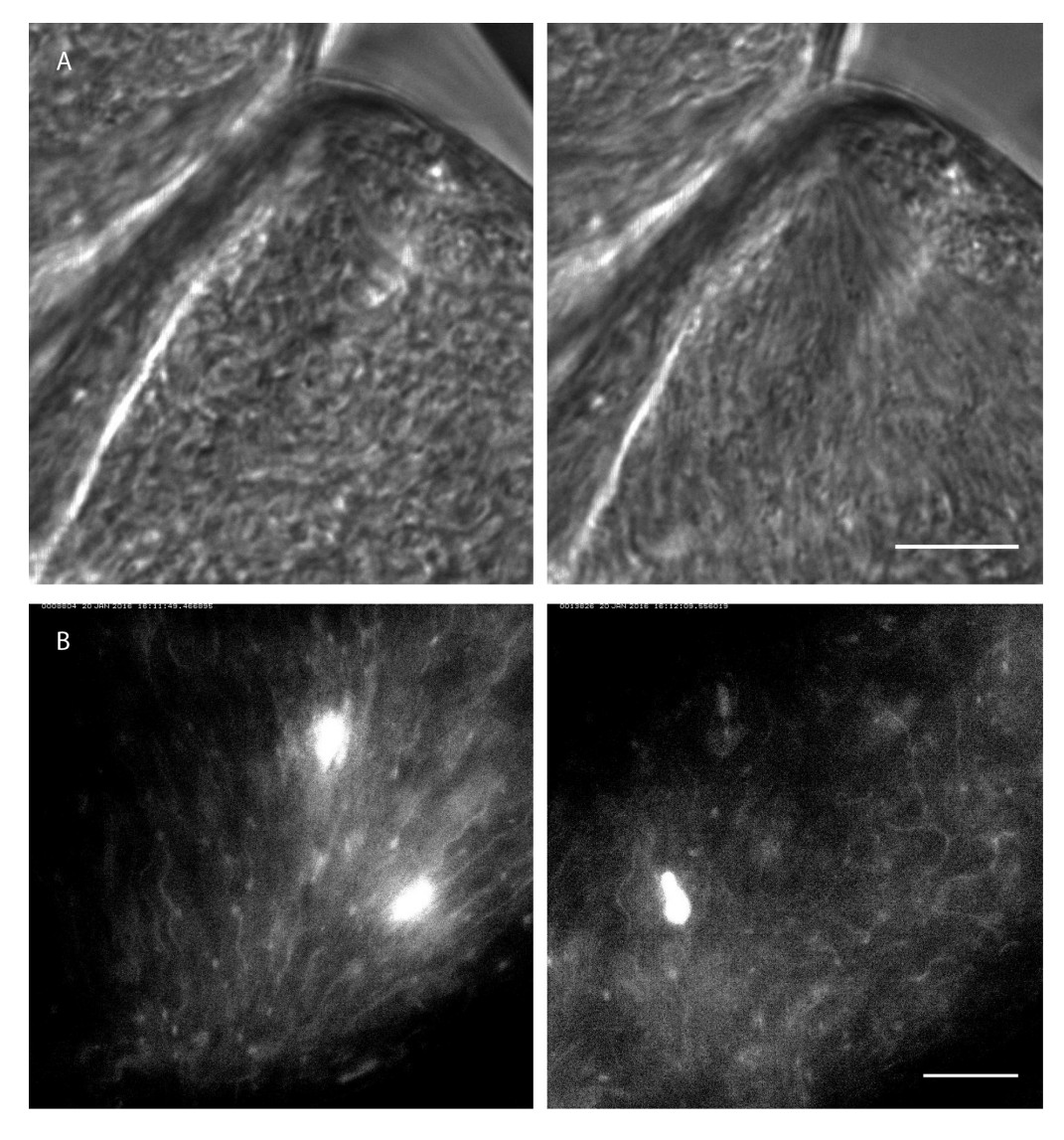

**Figure 10.** Rapid switching between synchronised and chaotic motion inside the ectoperitrophic space of the tsetse fly midgut. Infected tsetse fly midguts were dissected and analysed in PBS. Swarms of trypanosomes in the long procyclic to mesocyclic transition stages were recorded with 250 fps in the ectoperitrophic space of the anterior midgut (*Video 10*). Scale bars: 10 μm. (**A**) Switch from chaotic (left panel) to synchronised motion (right panel). (**B**) Synchronised motion (left panel) of cells and transition to chaotic movement (right panel) within a few seconds. Fluorescent trypanosomes express GFP in the nucleus and/or the flagellum.

Live high-speed imaging of trypanosome stages in the tsetse digestive tract completes the third and final task of our study, namely to gain an overview of the parasites motile behaviour, especially of the transitions between solitary and collective motion. The ability of mesocyclic forms for near-wall swimming and to form huge and dense clusters or swarms, without apparent attachments, is a distinctive feature, indicating the capacity for hydrodynamic self-organisation. Our study thus paves the way for powerful systems biophysics approaches to unravel the so far largely neglected interactions between microswimmers and their natural microenvironment.

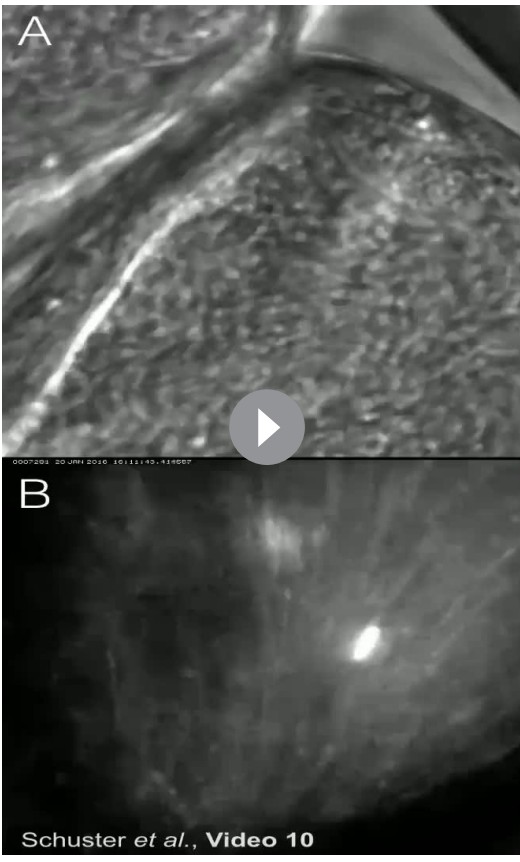

**Video 10.** Rapid switching between synchronised and chaotic motion inside the ectoperitrophic space of the tsetse fly midgut.

## Discussion

It is a still comparatively neglected possibility that the performance of microbial swimmers could be shaped by their microenvironment, very much in the same way as is the natural swimming behaviour of large animals. Nobody would doubt that the swimming motion and shape of tuna and carp are different because of the distinct ecological niches they occupy. Understanding microswimmer behaviour is less straightforward, mainly because we lack defined natural model environments for motion at low Reynolds numbers. The extended and developmentally programmed succession of microswimmers in different body parts of a small-sized insect like the tsetse fly provides such a tractable model.

Here, we introduce a complete set of methods, approaches and concepts that, in combination, will allow to elucidate what physical and biological constraints the versatile trypanosome microswimmers experience in the vast and multifaceted environment of their insect host.

In a first step, we decided to measure the topography of the tsetse interior. Multicolour LSFM allowed us to three-dimensionally reconstruct basically any region of the fly habitat with a resolution sufficient to visualise individual fluorescent trypanosomes, concomitant with the mapping of tissue structures in intact organs. LSFM produced astonishing views of the interior organisation of the fly gut, revealing the true topology of the PM to be extremely convoluted. This distension of the PM upon feeding and the extensive folding after excretion of liquid has been described elegantly in early work on *Glossina* (*Wigglesworth, 1929*). Incomprehensibly, the following publications on the PM did not appreciate the complexity of this environment. Studies were undertaken to establish the composition and the role of the PM in trypanosome infection, especially addressing the question how and where the parasites manage to cross this barrier, but data was focussed on detail regions of the matrix, mostly displaying single folds and necessarily in 2D (e.g. *Hoare, 1931*; *Yorke et al., 1933*; *Willett, 1966*; *Moloo et al., 1970*; *Ellis and Evans, 1977*; *Gibson and Bailey, 2003*).

We generated a high-resolution, multi-colour, three-dimensional map of the complete PM and the trypanosomes therein. The LSFM reconstructions also enabled the assignment of the two-dimensional structures observed by live microscopy of the dissected organs. The topology of folds and sheets of the PM seen in top view (i.e. *Figures 7C* and *8C*) could thus be compared with the 3D-views to allow a better understanding of the smooth confining structures the trypanosomes were swimming in and along (see below). Importantly, the spatial information gathered by LSFM yielded a quantitative assessment of the confinement states the parasites experience. Solitary swimmers were wrapped in single folded sheets, while co-resident collective assemblies of various sizes were identified between the PM sheets or between PM and the gut tissue. The precise three-dimensional maps allow mathematic analyses of the degrees of confinement and connectivity within the PM maze. The resultant data forms the basis for numerical simulations of single and collective swimming in this natural system of microchannels and crevices. In the living tsetse fly, the PM is not static but is dynamically folded and expanded during the digestion process. Although this initially means an impediment to microscopic analysis, LSFM approaches for rapid live microscopy will be available

(*Heddleston and Chew, 2016*) and the ability to record living microswimmers in such a complex, dynamic system will be an asset.

The LSFM measurements also allowed instructive visualisations of the intact, parasite-infested proventriculus (*Figure 4B*). The localisation of the proventricular trypanosome population revealed marked parasite concentration gradients along intact internal regions of the proventriculus. Again, these data extend the conventional views of trypanosome infestation in these organs (e.g. *Yorke et al., 1933*; *Gibson and Bailey, 2003*).

For live cell analysis by high-speed differential interference contrast (DIC) microscopy, the flies were cautiously dissected, without destroying the basic inner organisation of the trypanosome infested tissues. We frequently observed single parasites swimming along the extended walls of the PM. Microbe-surface interactions attract considerable attention because of their importance for sperm cells and bacteria (*Berke et al., 2008*; *Elgeti and Gompper, 2016*; *et al., 1963*). Sperm demonstrate a spiralling swimming trajectory that is modified by hydrodynamic and steric interactions at boundaries, resulting in rheotactic movement (*Kantsler et al., 2014*) or surface slithering, which might be beneficial under certain microenvironmental conditions (*Nosrati et al., 2015*). In contrast, the procyclic to mesocyclic trypanosome morphotypes appear to be adapted to swimming forwards in straight trajectories, which hardly change through contact with the PM. Near-wall motion could rather be guiding the cells by influencing the persistence of forward movement, as the trypanosomes can readily reverse the direction of flagellar waves and perform turning manoeuvres (e.g. *Video 7C*). If space is further restricted, the trypanosomes either reverse flagellar waves and back out of the confined space, or they perform sharp turns of the elastic cell body in order to leave the space by the default forward modus (e.g. *Videos 8A–C* and *9C*). All in all, trypanosomes arguably have a higher degree of control in navigating along walls and through confined spaces than sperm or bacteria.

When trypanosomes reach dead ends, the flagellum continues beating with constant frequency, and synchronisation of flagellar oscillations can be observed in groups of just few cells (*Figure 8A–C*, *Video 8A–C*) or in massive parasite clusters (*Figures 8D*, *9* and *10*, *Videos 8D*, *9* and *10*). Hydrodynamic synchronisation is a well-known phenomenon between sperm flagella ( *et al., 1963*; *Yang et al., 2008*) and leads to higher order swarm behaviours (*Elgeti and Gompper, 2016*; *Immler et al., 2007*; *Riedel et al., 2005*). The tsetse system allows the observation of all grades of collective accumulation, as distinct regions of the fly gut harbour different parasite concentrations. The propensity to synchronise is dependent on specific developmental forms, allowing the study of variations on swarming behaviour with changing morphologies. The trypanosomes add a further feature, as their reverse swimming ability allows single swimmers to leave the collectives. Parasite assemblies of identical cell densities can exhibit synchronised swimming behaviour or nematic organisation. They can be located in immediate vicinity, often in a single field of view. In the same area, spaces with freely swimming individual cells are detectable (*Figure 9*). We have generated fluorescent parasites, in order to identify single trypanosomes in densely populated organs and tissues of the fly and demonstrate possibilities for quantitative tracking of cell nuclei and individual flagella, resolving cell motility to single flagellar beats.

Thus, using the tsetse-trypanosome system, we now have the so far unique ability to track single microswimmers of distinct developmentally regulated morphotypes within their natural microenvironments. The natural habitat poses an important challenge though, as it will be necessary to elucidate the relevant physiochemical states prevailing for the parasite's developmental progression, in order to eventually establish complementary in vitro studies. As this system has naturally evolved, it is per definition complex and needs to be partly abstracted or deconstructed in order to challenge results with adjustable experimental parameters. For this purpose microfluidic systems have been established, especially for the analysis of bacteria (*Wu and Dekker, 2016*) and sperm research (*Knowlton et al., 2015*). Used mainly for biotechnological applications, few studies have focussed on microswimming and the elucidation of natural environmental factors (*Hochstetter and Pfohl, 2016*; *Stellamanns et al., 2014*; *Tung et al., 2015*; *Uppaluri et al., 2012*). A major approach to simulating natural environments is seen in research on artificial microswimmers, where both the motile components and the confinement conditions can be controlled (*Katuri et al., 2016*).

The tsetse system can efficiently bridge in vivo analyses and artificial in vitro systems. We have shown that many aspects of the natural environment can be described with sufficient precision to be mimicked by microfluidic structures. Furthermore, we can isolate any natural developmental stage

directly from the tsetse fly and study it in vitro, directly following in vivo analysis. We have detailed on the three-dimensional morphologies of all microswimmer types and their dynamic cellular waveforms, which gives us a realistic assessment of the cellular preconditions for distinctive motile behaviour. This data is not only invaluable for deducing the swimming mechanism of a cell in a certain hydrodynamic or confined environment, including interactions with neighbouring cells. It also allows the development of quantitative mathematic models, as we have previously shown in advanced numerical simulations of BSF trypanosomes (*Alizadehrad et al., 2015*). The predictive power, especially of multi-particle collision dynamics simulations, is underlined by the fact that the experimentally measured morphology of the mesocyclic stage agrees with the earlier in silico model (*Alizadehrad et al., 2015*): the angle of the helical flagellar course around the mesocyclic cell body was predicted to be shallower than that of the BSF (or the procyclic form, compare *Figure 5C* with *Figure 5A or I*).

The overall aim of this work was to introduce a tractable in vivo microswimmer system and demonstrate the resolution and the precision of the methods available for studies therein. Due to the broad scope of the work, the quantitative data is necessarily exemplary. For most future analyses, reference frames need to be defined, for example in the just mentioned detailed examination of the helix angle of the trypanosome´s attached flagellum, which constitutes a screw-like body, the conformation of the elastic cell body needs to described in a local, dynamic coordinate system. The dynamic characterisation of body axes in such a system also directly impacts the evaluation of motility-dependence on the flexibility of the cells interior microtubule corset. The interplay between cell body and flagellum generally needs to be taken into account for morphometry, for example in the search of an elusive resting state.

Another reference system to be described for quantitative microswimmer analysis, is the immediate vicinity of the trypanosome, combining the physical parameters of viscosity and flow of the surrounding fluid, as well as the degree and topology of confinement. Therefore, a straightforward parameter like median swimming speed turned out to be not instructive, instead we regard the maximum speed of a given morphotype as the most valuable parameter for assessing swimming capabilities (*Figure 6*). Surprisingly, the flagellar beat frequencies were relatively constant in most cells of one type, suggesting that the specific morphology of the cell plays a role in defining basic parameters of flagellar beating. The physical factors of the surrounding environment then impact on this beating pattern and determine absolute speed and direction of movement. These results are in agreement with those obtained in the analysis of different trypanosome species (*Bargul et al., 2016*; *Krüger and Engstler, 2016*), underlining the general relevance of our observations.

The results also have implications for fundamental models of flagellar beating. The suggested impact of the cell body´s properties on flagellar beating is consistent with a model of axonemal wave propagation control by mechanical bending forces, that is the geometric clutch hypothesis (*Lindemann and Lesich, 2015*; *Lindemann, 1994a*, *1994b*). In this model, dynein activity, causative of the curvature of flagella and cilia, is primarily controlled by mechanical forces acting transverse to the longitudinal arrangement of microtubules. In the case of the trypanosome flagellum, the attachment to the cell body would provide a permanent mechanical resistant force controlling the basic flagellar waveform. On the other hand, the mechanical forces exerted by the environment should also have a direct impact on beat regulation, which we are primed to quantitatively analyse in microfluidic systems.

In fact, such analyses with tsetse fly trypanosomes directly follow already mentioned experimental setups used to assess the reaction of mammalian trypanosomes to mechanical forces (*Heddergott et al., 2012*; *Bargul et al., 2016*). Here, micropillar arrays were designed to mimic the microenvironment of the bloodstream and high-speed, single cell analysis allowed the direct observation of the trypanosomes behaviour upon mechanical resistance. The design of such microfluidic devices is readily adapted, in order to create topologies mimicking the microenvironments described in this work. Thus, hypotheses on hydrodynamically controlled behaviour in certain compartments in the fly, for example, can be directly tested in the corresponding nature-inspired microflow chamber. We envisage producing such structures with elastic materials, which would allow the controlled application of forces on a confined population of parasites. In this way, interesting collective behaviour like the observed switching into synchronised flocks could be analysed.

In the next step, the microfluidic devices would be connected to pump systems, in order to evaluate the behaviour of the parasites to hydrodynamic flow. Controlled flow regimes would be

compared with in vivo measurements, using fluorescent beads together with fluorescent cell lines in tracking experiments (*Figure 9*, *Video 9*). Using the combination of flexible microfluidic devices and reversible microfluidic pump systems, even the analysis of peristaltic effects seems in reach.

Finally, the analysis of microswimmers in a biological system should be conducive to the elucidation of the patho-functional relevance of motile behaviour of specific parasite types. Motility data will help explain unresolved development processes, for example the journey of epimastigotes to the salivary glands or the crossing of the PM. Here, we have focused on the procyclic to mesocyclic stages, as these morphotypes showed prominent synchronisation of flagellar beats and spontaneously formed clusters. The collective microswimmers produce significant forces, capable of moving tissues, thus raising the questions of what influence the amazing numbers of parasites accumulating in the fly have on the surrounding microenvironment and on the trapped cells themselves.

## Materials and methods

### Trypanosome strain und culture

For this study the pleomorphic trypanosome strain EATRO 1125 (serodeme AnTat1.1) (*Le Ray et al., 1977*) was used. Procyclic cells were cultured at 27°C in SDM79 medium (*Brun and Schönenberger, 1979*) supplemented with 10% foetal bovine serum (*Hirumi and Hirumi, 1989*) and 20 mM glycerol (*Vassella et al., 2000*). Trypanosomes were used as wild type cells or transfected with 10 µg of linearised plasmid DNA using the AMAXA Nucleofector II (Lonza, Basel, Switzerland). To generate cells with fluorescent flagella the plasmid pPC PFR nt EGFP PFRAtag (*Adhiambo et al., 2009*), GeneID, PFR2: Tb927.8.4970) was used. For the fluorescently labelled cell nucleus a construct was used containing the plasmid pHD67E (*Bingle et al., 2001*) as backbone and the GFP sequence exchanged with the insert NLS:GFP from the plasmid p4231 (Sunter and Carrington, unpublished). To obtain the red fluorescent parasite strain, the GFP sequence of pHD67E was replaced with the tdTomato sequence.

### Tsetse fly maintenance and infection

Male and female flies of *Glossina morsitans morsitans* were used in this study. Flies were maintained in Roubaud cages at 27°C and 70% air humidity and fed with preheated defibrinated sheep blood through a silicon membrane three times a week.

For infections, teneral flies were fed trypanosomes in culture medium with their first meal 12–72 hr post-eclosion. For each infection we used around $1 \times 10^7$ cells/ml supplemented with 60 mM N-acetylglucosamine (*Peacock et al., 2006*).

For PM staining flies were fed with 40 µg/ml of WGA labelled with rhodamine (Vector Laboratories, Burlingame, CA) in preheated SDM79 medium.

### Tissue preparation for LSFM analysis

The samples were prepared for LSFM by modified protocols of previously described procedures (*Brede et al., 2012*; *Smolla et al., 2014*).

Flies were fixed in 4% PFA in PBS, after being numbed at 4°C and removal of extremities and the head. Flies were incubated for at least 2 days and stored at 4°C until further use. All solutions were replaced with transfer pipettes within the same glass vial to prevent injuries of the tissue. Flies were washed in PBS (2 × 10 min) and transferred into an aqueous solution of 30% hydrogen-peroxide (Sigma-Aldrich, St. Louis, MO) for 7–8 days. After bleaching, the flies were washed in PBS (3 × 10 min) and dehydrated in a graded ethanol series (30%, 50%, 70%, 80%, 90%, 2 hr each; 100% 12 hr minimum). Ethanol was replaced by n-Hexane (Sigma-Aldrich, St. Louis, MO) and incubated for 2 hr. The solution was exchanged stepwise with the clearing solution. The clearing solution (BABB) consists of 1 part benzyl alcohol (Sigma-Aldrich, St. Louis, MO) and two parts benzyl benzoate (Sigma-Aldrich, St. Louis, MO). After an incubation time of at least 2 hr at room temperature, tissues became optically transparent and suitable for imaging.

For the preparation of the digestive tract, flies were starved between 4–48 hr before dissection. The entire alimentary tracts, including the proventriculus and the midgut up to the Malpighian tubules, were surgically removed in a drop of PBS. Fresh tissue was immediately placed in 4% PFA in PBS, fixed for at least 24 hr and stored at 4°C until further use. Bleaching is not necessary for

alimentary tracts. The clearing procedure was performed as described above, except for a shorter n-Hexane treatment of 30 min.

### Image acquisition of tsetse fly tissue and processing of 3D models

We used a non-commercial fluorescence light sheet microscope equipped with a 5x and 20x objective for imaging the digestive systems (*Brede et al., 2012*) and a microscope from LaVision for imaging of the fly body. Image stacks were recorded in 2 µm steps. All settings were managed with the Andor iQ 2.9.1 Software (Andor Technology Ltd, Belfast, UK). Three-dimensional models were created using Amira software package v6.1.1 (FEI, Munich, Germany).

### Parasite density scoring and analysis

Batches of 4 to 11 infected tsetse flies were dissected daily up to 25 days after ingestion of the infective meal (n = 196 infected flies). The digestive tracts were isolated and stretched from the proventriculus to the posterior midgut in a drop of PBS for observation under a SMZU epifluorescence stereomicroscope (Nikon) with a CoolPix 950 camera (Kodak). Infected samples were further scrutinised under a DMI4000 microscope (Leica) and images were acquired with a Retiga-SRV camera (Q39 Imaging). The digestive tract was virtually segmented into 17 distinct zones corresponding to the number of microscopic fields required to cover its entire length (six for the posterior midgut, 10 for the anterior midgut and one for the PV). The number of fluorescent parasites in each zone was counted and scaled with an arbitrary 4-level colour scheme. The mean density and post-infection time-point in each zone were cumulatively plotted.

### Trypanosome morphometry

Tsetse flies were starved for at least 48 hr and dissected in a drop of PBS. The digestive system, including the proventriculus and the salivary glands, of 2–3 positive flies were cut into small pieces using a razor blade allowing the trypanosomes to float out of the tissue. The tissue was removed and the parasite cell-surface was labelled with 1 mM Atto488-NHS (Atto-tec GmbH, Siegen, Germany) for 10 min. Incubation was carried out at 4°C in the dark. Cells were fixed in 4% PFA and 0,25% glutaraldehyde in PBS buffer overnight at 4°C and washed at 500 g for 5 min in PBS. Cells were stained with 0,5 mg/ml DAPI.

Image stacks of trypanosomes were recorded with a fully automated wide field fluorescence microscope iMIC (FEI, Munich, Germany), controlled by the Live acquisition software v2.6.0.19 (FEI, Munich, Germany) and equipped with a 100x objective. Stacks were generated with 100 slices and 100 nm step size and deconvolved using the Huygens Essential Image software package v16 (SVI, Hilversum, Netherlands). 3D models were computed with the Amira software v6.1.1 (FEI, Munich, Germany) using an edge detection filter (Sobel) and volume models using the Voltex display function. Flagella were traced using the volume model and Amira´s filament editor tool.

The developmental stages of the recorded cells were identified according to the documented two-dimensional parameters of cell length, width, flagellar length and relative positions of nuclei and kinetoplasts (*Rotureau et al., 2011*, *2012*; *Sharma et al., 2008*; *Subota et al., 2011*; *Van Den Abbeele et al., 1999*).

### Video acquisition and analysis of trypanosome motility

Tsetse flies were dissected at different infection time points and used for video acquisition. For high-speed analysis the whole alimentary tract including the salivary glands was dissected and spread lengthwise on a glass slide in PBS and covered with a cover slip. Live cell microscopy was performed at room temperature, with an inverted fully automated DMI6000 wide-field microscope (Leica Microsystems, Mannheim, Germany), equipped with a 100x oil and a 63x glycerol objective. Parasites were recorded inside the intact issue or were expelled from the tissue and released in the surrounding PBS. For high-speed recording the sCMOS camera pco.edge (PCO, Kelheim, Germany) was used at frame-rates of 100–250 fps. The total duration of imaging did not exceed 30 min.

For single cell analysis selected sequences were processed with Fiji or Amira. The swimming speed was calculated after measuring the covered distance in the direction of the cell´s movement as described previously (*Bargul et al., 2016*).

For cellular waveform analysis we choose representative videos of trypanosomes isolated from the fly and swimming persistently forward with uninterrupted tip-to-base beats. Date processing was performed as described previously (*Bargul et al., 2016*). Cell tracking was performed using the Imaris Software package v8 (Bitplane, Zürich, Switzerland).

## Additional information

### Funding

| Funder | Grant reference number | Author |
|---|---|---|
| Deutsche Forschungsgemeinschaft | EN 305 | Markus Engstler |
| Deutsche Forschungsgemeinschaft | SPP 1726 | Markus Engstler |
| Deutsche Forschungsgemeinschaft | GRK 2157 | Markus Engstler |

The funders had no role in study design, data collection and interpretation, or the decision to submit the work for publication.

### Author contributions

SS, Formal analysis, Investigation, Visualization, Methodology, Writing—original draft, Writing—review and editing; TK, Conceptualization, Data curation, Formal analysis, Supervision, Validation, Investigation, Visualization, Methodology, Writing—original draft, Writing—review and editing; IS, Supervision, Investigation, Methodology; ST, BR, Investigation, Methodology; AB, Resources, Methodology; ME, Conceptualization, Data curation, Formal analysis, Supervision, Funding acquisition, Validation, Investigation, Visualization, Methodology, Writing—original draft, Project administration, Writing—review and editing

### Author ORCIDs

Timothy Krüger, http://orcid.org/0000-0001-5544-1098
Markus Engstler, http://orcid.org/0000-0003-1436-5759

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
