## [Decision Letter]

Thank you for submitting your article "Developmental adaptations of trypanosome
motility to the tsetse fly host unravel a capacious in vivo microswimmer system"
for consideration by *eLife*. Your article has been reviewed by three
peer reviewers, and the evaluation has been overseen by a Reviewing Editor and Ian
Baldwin as the Senior Editor. The following individual involved in review of your
submission has agreed to reveal his identity: Charles Lindemann (Reviewer #3).

The reviewers have discussed the reviews with one another and the Reviewing Editor has
drafted this decision to help you prepare a revised submission.

Summary:

The authors employed cutting-edge light microscopy methods including multicolour light
sheet fluorescence microscopy and high speed video microscopy to provide a detailed
description of the Tsetse fly alimentary tract tissue topology. They went on and
scrutinized the cell morphology and swimming behaviours of the different life cycle
stages of the Trypanosoma brucei parasites that navigate through these natural
microenvironments. The "tour de force" of this work resides in the
unprecedented 3D reconstruction of the host digestive system and visualization of the
location of parasites as infection progresses through the morphing stages of the
parasite and through the anatomy of the host at an unprecedented temporal and spatial
resolution.

This system enables important future hypothesis-driven research ranging from the basic
biology of an important parasite to systems-level description of micro swimmer
behaviours.

Essential revisions:

1) The Introduction and the Discussion are heavily skewed to making a case that the
importance of the work is to provide a platform or model system for computational
biology of microswimmers. The microenvironments and the complex morphology of this
system do not readily lend themselves to the simplification that is typically needed for
mathematical modeling. This is not considered as a system of choice given the numerous
variables that difficult to control and hard to simulate. The authors would do better to
focus both introduction and discussion on the novel findings of their own report, which
are considerable.

2) The study revealed interesting patterns of solitary and collective motion and reports
rapid switching between synchronized and "chaotic" motion of cells in the
ectoperitrophic space of the midgut. On this latter point, the authors raise the
question whether synchronization of cell behaviour was driven by chemical cues or by
hydrodynamic self-organisation. The data presented in Figure 10 is interpreted as "unambiguous" (subsection
“Self-organisation of parasites by hydrodynamic interaction”, fourth paragraph) evidence
for the latter. The basis for this conclusion is not entirely clear. The authors say a
chemical process would "generally be slower" (first paragraph of the
aforementioned subsection) and it would be helpful if they could be more precise about
the difference in time scales so that the cell behaviour in Video 10 can be assessed against these expectations.

3) Despite the stunning quality of the images and videos produced the study does not
actually reveal anything new about the tsetse-trypanosome interaction. Even the
"astonishing degree of convolution" of the peritrophic matrix (PM) (subsection
“Multicolour light sheet fluorescence microscopy reveals the complex three-dimensional
architecture of the microswimmer habitats in the tsetse vector”, third paragraph and
Discussion, third paragraph) has been described previously, including data on its
dimensions, as it collapses when water from the bloodmeal is excreted. The claim for
priority is unfounded: "This was a surprising finding, which allowed a first
realistic view into an important environmental compartment for the procyclic and
mesocyclic stages of *T. brucei*." Entomologists in the last century
were well aware of this e.g. Wigglesworth (1929). The authors are referring to cartoons
of the PM: "a simple hollow sleeve as mostly depicted in the literature".
Incidentally, the description of the PM as "a chitinous sleeve" is also
inaccurate, as it contains a lot of protein as well as chitin. A comprehensive coverage
of the literature is a prerequisite.

4) The authors justify the study as a prelude to investigating "what physical and
biological constraints the versatile trypanosome microswimmers experience in the vast
and multifaceted environment of their insect host", but it is not clear what
hypotheses they would test in future studies. They claim that the tsetse
microenvironment offers a tractable model system for studying microswimmers, but the
tsetse milieu is not static, changing both in terms of its own movements – peristalsis
of the alimentary tract and salivary glands – and in terms of the chemical environment
as the infected bloodmeal is digested and further bloodmeals are taken. Comparison with
controlled microfluidic environments may be more informative, e.g. to explore why
trypanosomes change from synchronized to chaotic swimming (Video 10).

---

## [Author Response]

*Essential revisions:*

*1) The Introduction and the Discussion are heavily skewed to making a case that
the importance of the work is to provide a platform or model system for computational
biology of microswimmers. The microenvironments and the complex morphology of this
system do not readily lend themselves to the simplification that is typically needed
for mathematical modeling. This is not considered as a system of choice given the
numerous variables that difficult to control and hard to simulate. The authors would
do better to focus both introduction and discussion on the novel findings of their
own report, which are considerable.*

We agree with the reviewers criticism of overly promoting the model system for
computational biology in parts of Introduction and Discussion. Therefore, we have now
removed the redundant mentions of the model character of our work and focus more on the
biological impact. Nevertheless, the intention of the study was primarily to introduce
the tsetse vector as an enclosed and tractable system as a whole, for studying the
natural complexity of microswimmer interactions with/in highly intricate environments.
It is true that today the system is by far too little understood to allow mathematical
modelling in its entirety. But we can learn from the fly. For example, we have been able
to model the swimming behaviour of the different morphotypes, and we have identified in
vivo motion patterns that can be modelled, e.g., the near-wall swimming of mesocyclic
parasites along the peritrophic membrane (see also our reply to comment 4).

*2) The study revealed interesting patterns of solitary and collective motion and
reports rapid switching between synchronized and "chaotic" motion of cells
in the ectoperitrophic space of the midgut. On this latter point, the authors raise
the question whether synchronization of cell behaviour was driven by chemical cues or
by hydrodynamic self-organisation. The data presented in Figure 10 is interpreted as "unambiguous" (subsection
“Self-organisation of parasites by hydrodynamic interaction”, fourth paragraph)
evidence for the latter. The basis for this conclusion is not entirely clear. The
authors say a chemical process would "generally be slower" (first paragraph
of the aforementioned subsection) and it would be helpful if they could be more
precise about the difference in time scales so that the cell behaviour in Video 10 can be assessed against these
expectations.*

Indeed, the text was imprecise; “unambiguous” was referring to the fact that there is no
evidence of cell attachment and that hydrodynamic interactions are most likely. We have
rephrased and clarified the paragraph, including the direct comparison to the very
limited data there is available regarding time scales of potential chemotactic behaviour
in trypanosomes:

“It should be noted, that the timescale of a few seconds observed here for collective
organisational status change is significantly smaller than the shortest known timescale
of potential chemotactic behaviour in trypanosomes, which is at least several minutes
(Oberholzer et al., 2010). Although chemical signals will undoubtedly be relevant for
rapid fluctuations of flagellar beating, an adaptive reaction system required for
directed behaviour of parasite swarms is unlikely to be responsible for establishing the
switching behaviour observed in our experiments.”

*3) Despite the stunning quality of the images and videos produced the study does
not actually reveal anything new about the tsetse-trypanosome interaction. Even the
"astonishing degree of convolution" of the peritrophic matrix (PM)
(subsection “Multicolour light sheet fluorescence microscopy reveals the complex
three-dimensional architecture of the microswimmer habitats in the tsetse vector”,
third paragraph and Discussion, third paragraph) has been described previously,
including data on its dimensions, as it collapses when water from the bloodmeal is
excreted. The claim for priority is unfounded: "This was a surprising finding,
which allowed a first realistic view into an important environmental compartment for
the procyclic and mesocyclic stages of T. brucei." Entomologists in the last
century were well aware of this e.g. Wigglesworth (1929). The authors are referring
to cartoons of the PM: "a simple hollow sleeve as mostly depicted in the
literature". Incidentally, the description of the PM as "a chitinous
sleeve" is also inaccurate, as it contains a lot of protein as well as chitin. A
comprehensive coverage of the literature is a prerequisite.*

We appreciate this important comment, especially the Wigglesworth reference, which we
have now cited. Furthermore, we have screened the literature for more information on the
convolution of the PM, however, could not find better data than provided by Wigglesworth
(1929) and maybe Yorke (1933). In fact, most publications describe the PM only as a
continuous, hollow sleeve or tube. We have added a paragraph in the Discussion with the
most relevant citations:

“This distension of the PM upon feeding and the extensive folding after excretion of
liquid has been described elegantly in early work on Glossina (Wigglesworth, 1929). […]
Studies were undertaken to establish the composition and the role of the PM in
trypanosome infection, especially addressing the question how and where the parasites
manage to cross this barrier, but data was focussed on detail regions of the matrix,
mostly displaying single folds and necessarily in 2D (e.g. Hoare, 1931; Yorke et al.,
1933; Willett, 1966; Moloo et al., 1970; Ellis and Evans, 1977; Gibson and Bailey,
2003).”

We have rephrased "a chitinous sleeve" to:

“non-cellular, glycosaminoglycan, glycoprotein and chitin containing, cylindrical
sleeve”

Our work provides, among other things, a detailed 3D-topological map of the geometry of
the convoluted PM. We show the distribution of the parasites in the folds and we reveal
the motion pattern of the trypanosomes in their natural environment. We would like to
believe these aspects and the combination thereof are of considerable novelty.

*4) The authors justify the study as a prelude to investigating "what
physical and biological constraints the versatile trypanosome microswimmers
experience in the vast and multifaceted environment of their insect host", but
it is not clear what hypotheses they would test in future studies. They claim that
the tsetse microenvironment offers a tractable model system for studying
microswimmers, but the tsetse milieu is not static, changing both in terms of its own
movements – peristalsis of the alimentary tract and salivary glands – and in terms of
the chemical environment as the infected bloodmeal is digested and further bloodmeals
are taken. Comparison with controlled microfluidic environments may be more
informative, e.g. to explore why trypanosomes change from synchronized to chaotic
swimming (Video 10).*

As stated above our study marks a starting point. It provides a static blueprint of the
tsetse topology and the distribution of the different trypanosome morphotypes. In the
next step, we e.g. explore fluid flow at different scales, ranging from rather slow
microflows caused by the trypanosomes itself to massive flows produced by peristalsis.
In fact, our current and future work is nature-inspired, i.e. we measure the real
environments in the fly and the behaviour of the trypanosomes therein. Then, we abstract
from nature and construct microfluidic environments that allow the control of
parameters. The results from these experiments could provide hints that might help to
explain the in vivo behaviour of the trypanosome microswimmers. We now provide more
information of this workflow in the Discussion:

“In fact, such analyses with tsetse fly trypanosomes directly follow already mentioned
experimental setups used to assess the reaction of mammalian trypanosomes to mechanical
forces (Heddergott et al., 2012; Bargul et al., 2016). […] Using the combination of
flexible microfluidic devices and reversible microfluidic pump systems, even the
analysis of peristaltic effects seems in reach.”